behaviour

intergroup conflict, territorial intrusion, cooperation, dwarf mongooses, field experiment

**Author for correspondence:**
Amy Morris-Drake
e-mail: am9162@bristol.ac.uk

# Extended and cumulative effects of experimentally induced intergroup conflict in a cooperatively breeding mammal

Amy Morris-Drake[1], Jennifer F. Linden[1], Julie M. Kern[1,2] and Andrew N. Radford[1]

[1]School of Biological Sciences, University of Bristol, 24 Tyndall Avenue, Bristol BS8 1TQ, UK
[2]School of Environmental and Rural Science, University of New England, Armidale, New South Wales 2351, Australia

(iD) AM-D, 0000-0003-4243-4651; JMK, 0000-0002-7619-8653; ANR, 0000-0001-5470-3463

Conflict between rival groups is rife in nature. While recent work has begun exploring the behavioural consequences of this intergroup conflict, studies have primarily considered just the 1–2 h immediately after single interactions with rivals or their cues. Using a habituated population of wild dwarf mongooses (*Helogale parvula*), we conducted week-long manipulations to investigate longer-term impacts of intergroup conflict. Compared to a single presentation of control herbivore faeces, one rival-group faecal presentation (simulating a territorial intrusion) resulted in more within-group grooming the following day, beyond the likely period of conflict-induced stress. Repeated presentations of outsider cues led to further changes in baseline behaviour by the end of the week: compared to control weeks, mongooses spent less time foraging and foraged closer to their groupmates, even when there had been no recent simulated intrusion. Moreover, there was more baseline territorial scent-marking and a higher likelihood of group fissioning in intrusion weeks. Consequently, individuals gained less body mass at the end of weeks with repeated simulated intrusions. Our experimental findings provide evidence for longer-term, extended and cumulative, effects of an elevated intergroup threat, which may lead to fitness consequences and underpin this powerful selective pressure.

## 1. Introduction

In many social species, from ants to humans, groups are in conflict with conspecific outsiders over access to limited resources [1–3]. Single intruders or same-sex groups may attempt to monopolize reproductive opportunities or usurp dominant individuals, while whole groups may invade territories and aim to acquire space, food and sleeping sites [4–6]. An extensive literature exists on how animals behave during outgroup interactions, considering the type of encounter, who contributes during contests and the factors that influence the outcome [7–9]. Recently, the behavioural consequences of outgroup conflict have received increased research interest [10]. For instance, there is strong empirical evidence for alterations in within-group affiliation in response to outgroup conflict in a wide range of taxa, including mammals [11–13], birds [14,15], fish [16,17] and insects [18]. Exposure to outgroup threats has also been shown to influence other behaviours, such as group movement patterns and individual decisions about foraging and vigilance [12,19,20]. However, the majority of this work has focused on short-term effects (over minutes or hours) in the immediate aftermath of single interactions with outsiders or cues of their presence [12–17,20] (but see [21–23]). To understand fully the

effects of outgroup conflict, investigation is also needed of extended effects from single interactions and cumulative effects from repeated interactions.

Some short-term behavioural responses to outgroup conflict (e.g. changed movement patterns) probably arise as a result of temporary territorial exclusion and/or avoidance of conflict zones, and the subsequent reduced access to resources such as preferred foraging locations [13,19,20]. But there are indications from a few observational studies that resource-use decisions could continue to be affected even when spatial access is not restricted [24,25]. For instance, green woodhoopoe (Phoeniculus purpureus) groups that had engaged in an extended intergroup contest in the morning were more likely than on control days to roost in the encounter area in the evening, even if they had lost the contest [24]. Some additional behavioural changes in the immediate aftermath of outgroup conflicts are likely to be explained by an acute stress response, since interactions with conspecific outsiders can activate the hypothalamic–pituitary–adrenocortical (HPA) axis, leading to an increase in glucocorticoid (GC) hormone secretion [26–28]. For example, increased within-group affiliation could result because grooming causes a reduction in anxiety [27,29]. However, grooming may also fulfil a social function—for instance, as a reward for defensive efforts in previous contests or to promote participation in future encounters [14,15,30]—and thus increased within-group affiliation might persist even when contest-related elevations in stress levels have dissipated [24]. This might be especially true if, for example, there is an increased likelihood of another contest occurring soon. Experiments testing the lasting behavioural consequences of outgroup conflict are therefore needed to determine if there are extended effects from a single interaction once periods of spatial exclusion and elevated stress have ceased.

While it is logistically simplest for researchers to investigate responses to a single outgroup interaction, behavioural decisions are probably also influenced by prior events and the cumulative build-up of threat. As with other stressors [31], previous outgroup interactions might increase (e.g. through sensitization) or lessen (e.g. through habituation) the responses to a current conflict situation. A recent laboratory-based study on harvester ants (Messor barbarus), for example, found evidence for a 'priming' effect: if there had been an earlier presentation of an intruder (20 min before), there was a greater increase in ant activity and contact between groupmates in response to a second intrusion [18]. Multiple outgroup interactions could also have a cumulative effect; such a build-up of threat could result in behavioural changes not only in the immediate aftermath of each interaction, but also more generally [22]. From a proximate perspective, repeated exposure to a stressor could lead to dysregulation of the HPA axis, and consequent changes to baseline behaviour [32,33]. From an ultimate perspective, cumulative effects of outgroup conflict could lead to group members being more affiliative or cooperative, with increases in within-group grooming or a reduced likelihood of group fissions [24,34,35]. Similarly, a greater general threat level might result in higher investment in vigilance or defensive actions [12]. Two observational field studies have suggested a positive link between outgroup interaction frequency and within-group behaviour: green woodhoopoe and chimpanzee (Pan troglodytes) groups experiencing more intergroup conflict had higher rates of within-group affiliation and association [14,34]. However,

these results could arise because the occurrence of more intergroup interactions (IGIs) means that there are more post-conflict periods (when behaviour is known to change), rather than a more general change in baseline behaviour at times when there has been no recent interaction. To investigate fully the cumulative effects of outgroup conflict, we therefore need field experiments to determine behavioural changes outside of the immediate post-conflict period.

Inter-individual differences in responses to outgroup conflict are expected due to variation in benefits and costs. Groups are a heterogeneous mix of individuals who differ in, for example, age, sex and dominance status, and it is well understood that these attributes influence the incentive to engage in outgroup contests as the perceived threat and cost of participation is not the same for all group members [1,7,36]. Recent studies have demonstrated that individuals can also differ in within-group behavioural responses following outgroup interactions depending on their sex and dominance status. For example, captive experiments with cichlid fish (Neolamprologus pulcher) found that the intrusion of an outsider altered the levels of affiliation and aggression shown towards groupmates, but that these varied depending on the donor's own characteristics, as well as those of potential recipients and the identity of the intruder [16,17]. In a field experiment, post-contest increases in within-group affiliation by green woodhoopoes were the result of dominants grooming subordinates more [15]. To date, such studies have focused on within-group behaviour during or immediately after a single outgroup interaction; longer-lasting inter-individual differences resulting from single or repeated outgroup interactions have not been investigated.

Here, we conducted a field experiment with wild dwarf mongooses (Helogale parvula) to determine extended effects from a single simulated territorial intrusion by a rival group, cumulative effects after repeated simulated intrusions and inter-individual variation in responses. Dwarf mongooses are an ideal species for such a study because they can be habituated to the close presence of observers, allowing experimental manipulations and detailed monitoring in natural conditions [37,38]. They live in relatively stable, cooperatively breeding groups of up to 30 individuals, comprising a dominant breeding pair and nonbreeding subordinates of both sexes [37,39]. Group members cooperate to defend a shared territory from conspecific rivals [20,40]. IGIs range from signal exchanges (mainly visual and/or acoustic) to violent confrontations that can lead to serious injury (A.M.-D. 2021, personal observation). Recent work has revealed an array of short-term behavioural changes, including increased grooming and sentinel behaviour, and reduced nearest-neighbour foraging distances and group movement, in the hour after a simulated intergroup threat [12,20]. In the current study, we simulated multiple territorial intrusions by a rival group across several days and compared behaviours with control periods. In general, we expected this intergroup conflict to result in some extended effect on individual behaviour the day after a single simulated intrusion, but for there to be greater effects following repeated simulated intrusions during the week (e.g. for there to be increases in grooming and sentinel behaviour, and a reduction in foraging). We also expected the cumulative threat to affect group-level behaviour (e.g. for there to be an increase in territorial scent-marking and a reduction in the likelihood of groups

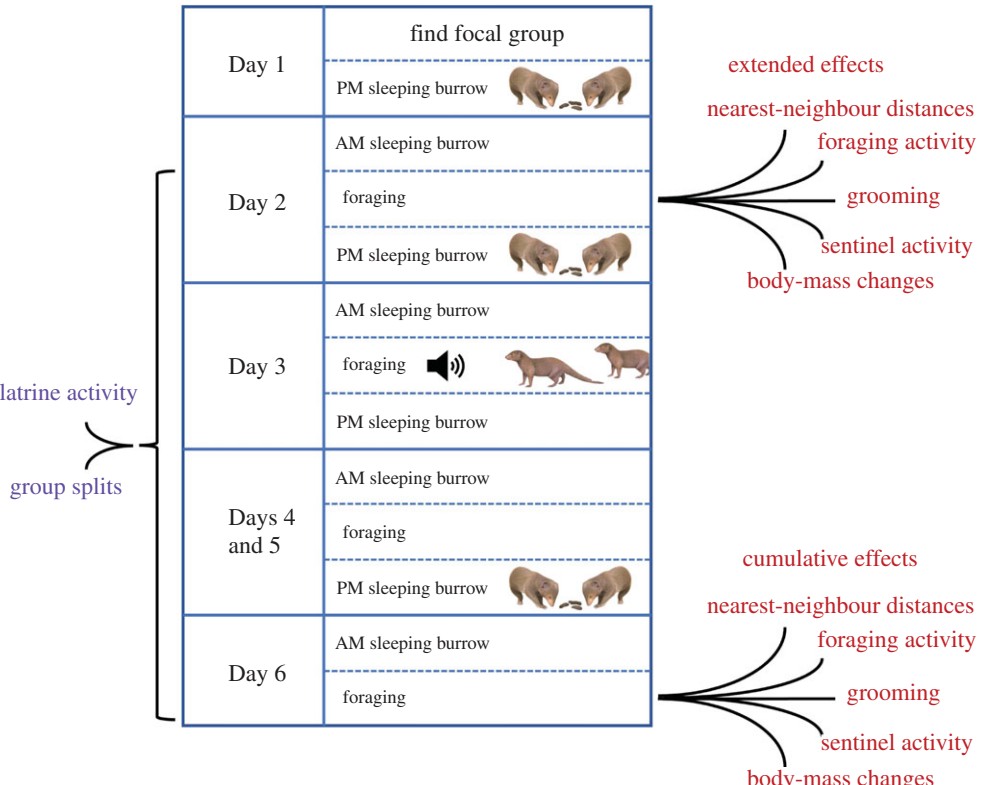

**Figure 1.** Illustration of a typical treatment week. In the experiment, each group (*n* = 7) received two week-long treatments: an Intrusion week where the faecal presentations and call playback simulated the presence of a rival group, and a Control week where herbivore faeces and calls were used on an equivalent schedule. Data collected on Day 2 were used to investigate the extended effects after the first simulated intrusion on Day 1, while data collected on Day 6 were used to investigate the cumulative effects after experiencing repeated simulated intrusions. Rare group-level behaviours were also collected over the course of the week. (Online version in colour.)

splitting up while foraging). We predicted that such changes in behaviour would result in a negative impact on body-mass gains by the end of trial weeks. In terms of inter-individual variation, we expected intergroup conflict to have a stronger effect on dominants than subordinates, because the former have the most to lose (breeding position, territory) if a rival group invaded [15], and on females compared to males, because the former are the philopatric sex in dwarf mongooses [40] and thus the retention or loss of contested resources could have longer-term consequences for females [41,42].

## 2. Methods

### (a) Study overview

Our field-based repeated-measures experiment entailed two treatments, each one-week long (figure 1), to each of seven habituated dwarf mongoose groups in counterbalanced order; full details of the Dwarf Mongoose Research Project study population in electronic supplementary material. During Days 1–5 of an Intrusion week, we presented the focal group with the simulated presence (faecal samples or call playback) of the same non-neighbouring rival group. Non-neighbouring groups, such as transients or newly established groups seeking a new territory, naturally encroach into the territory of other groups [43,44]. During a Control week, we presented the focal group with herbivore faeces and call playback on an equivalent schedule to the Intrusion week; herbivore stimuli have been used as controls in previous dwarf mongoose experimental studies examining the short-term consequences of single intergroup intrusions [12,20]. We conducted faecal presentations most evenings during treatment weeks (mean ± s.e.: 4.4 ± 0.2, range:

3–5), always at the sleeping burrow, while one playback was undertaken mid-week (Day 3 or 4) when the group was out foraging; the playback was for the purpose of maintaining rival group or herbivore exposure. We used data collected while the group was foraging away from the sleeping burrow on Day 2 (the day after the first simulated intrusion) to determine whether there were any extended effects of single intergroup events on individual behaviours (grooming, foraging, sentinel activity). We collected the same behavioural data on Day 6 (after repeated simulated intrusions) to investigate possible cumulative effects of intergroup threat. To examine overall treatment differences in relatively rare group-level behaviours, we also collected data on territorial scent-marking (latrining) and group fissions throughout the week. Finally, we used data from Days 2 and 6 to determine if the behavioural changes influenced daily body-mass gains after single and repeated simulated intrusions.

### (b) Simulated intrusions

The use of dwarf mongoose and herbivore faecal presentations followed the general protocol of Morris-Drake *et al.* [12]. For Intrusion weeks, we used fresh dwarf mongoose faeces from groups that did not share any territorial boundaries with the focal groups; for Control weeks, we used fresh giraffe (*Giraffa camelopardalis giraffe*) faecal pellets (similar in diameter to dwarf mongoose faeces). Full details of faecal collection and use are in the electronic supplementary material. In Intrusion weeks, each faecal presentation comprised one deposit from four different adult group members (of both sexes), including at least one dominant individual. Different faecal samples were used on each presentation day in both treatments.

The use of dwarf mongoose and herbivore call playbacks also followed the general protocol of Morris-Drake *et al.* [12]. We constructed playback stimuli from original sound recordings; full

details of sound recording and playback-track construction in the electronic supplementary material. For Intrusion weeks, we used playbacks of close and lost calls to simulate the nearby presence of a rival group. Close calls (low-pitched vocalizations) are produced continuously when dwarf mongooses are foraging and moving [37,45], while lost calls (high-pitched vocalizations) are given in a range of circumstances, including coordinating lost group members and group movement, as well as during IGIs [46,47]. We obtained close-call recordings from four adult group members (either one dominant and three subordinates or both dominants and two subordinates), and recorded lost calls from two adult group members. For a given focal group, audio recordings were made from the same non-neighbouring group from which faeces were collected. For Control weeks, we recorded zebra (*Equus quagga*) and blue wildebeest (*Connochaetes taurinus*) sounds from close to the main lodge at the study site. All playbacks were of natural call rates and at natural amplitudes. We constructed unique rival group and control tracks for different groups.

## (c) Experimental protocol

We counterbalanced treatment order between groups as much as possible: four groups received the Intrusion week first, while three received the Control week first. Our aim was to leave two weeks between treatments to the same group but we had to abandon and repeat three trials because required conditions were not met (mean ± s.e. days between treatments: 18.7 ± 3.5, range: 4–36; full details in electronic supplementary material). We conducted faecal presentations at the evening sleeping burrow; burrows are regularly contested between groups [43]. To reflect the natural situation whereby a rival group had deposited faeces at a burrow while the focal group had been away foraging, the observer moved ahead of the focal group and placed the relevant faecal sample near the sleeping burrow (full details in electronic supplementary material). We conducted one playback on either Day 3 or 4 of trial weeks (depending on which day had the best weather forecast) when the group was foraging (full details in electronic supplementary material).

## (d) Data collection

To verify that rival-group faecal presentations induced an intergroup response as planned, we recorded the immediate reactions to the first (Day 1) and last (Day 5) faecal presentation in each trial week; full details in electronic supplementary material. There was a significantly greater proportion of individuals interacting with the rival-group faeces (mean ± s.e. = 0.34 ± 0.06) compared to the control faeces (0.11 ± 0.04) on Day 1 (Wilcoxon signed-ranks test: $Z = 2.207$, $n = 7$, $p = 0.031$), although the effect was dampened slightly by Day 5 ($Z = 1.782$, $n = 7$, $p = 0.094$; rival: 0.37 ± 0.09, control: 0.09 ± 0.05). However, the total time spent sniffing the faeces was significantly longer for rival-group faeces than control faeces on both Day 1 ($Z = 2.366$, $n = 7$, $p = 0.016$; rival: 20.1 ± 6.0 s; control: 1.9 ± 0.7 s) and Day 5 ($Z = 2.197$, $n = 7$, $p = 0.031$; rival: 21.9 ± 7.1 s; control: 1.1 ± 0.6 s).

To assess behavioural changes the day after exposure to a single rival-group intrusion and after repeated intrusions, we conducted observations on Days 2 and 6 of each treatment week, following established Dwarf Mongoose Research Project protocols [48,49]; full details in electronic supplementary material. Within-group affiliation is one of the most commonly considered responses when assessing the immediate behavioural consequences of outgroup conflict [13,15,16], with affiliation displayed through allogrooming in dwarf mongooses [49]. So, we recorded ad libitum the duration and the identity of those individuals involved in grooming bouts between adults when the group was foraging away from the burrow during the day. Throughout the day, dwarf mongooses make constant decisions relating to foraging (e.g. how much time to spend foraging and how close to forage to groupmates [50]) and vigilance (e.g. whether to act as a sentinel [51]). We therefore conducted scan-samples every 15 min during the day to record whether the group was foraging, and scan-samples every 30 min to estimate the distance between foraging nearest-neighbours and to record whether a sentinel was present. Sentinels were defined as individuals actively scanning their surroundings from an elevated post (termite mound, rock, tree)—feet at least 10 cm off the ground—while their groupmates were engaged in other activities (usually foraging) [37,51].

By monitoring groups over the course of a whole trial week, we could also gain sufficient data to consider treatment effects on relatively sporadic group-level behaviours. One aspect of dwarf mongoose territorial defence is the depositing of scent-marks (urine, faeces, cheek-gland and anal-gland secretions) at communal latrines (rocks, trees, termite mounds) around their territory [20]. As group members sometimes leave the main foraging party to latrine (A.M.-D. 2021, personal observation), this can cause group fissions when the main group splits into subgroups. Throughout each trial week, we therefore collected data ad libitum on the occurrence and duration of any latrine events and noted the occurrence of group fissions.

In addition to the behavioural data, we weighed individuals for assessment of body mass on Day 2 and Day 6. Study individuals are trained to climb onto electronic weighing scales in exchange for a small amount of hard-boiled egg [48]. Adults and independently foraging pups were weighed after emergence at the morning sleeping burrow (before leaving to start foraging) and again after a 3 h foraging session, to determine body-mass changes.

## (e) Data analysis

All statistical tests were two-tailed and considered significant at $p < 0.05$. We conducted parametric tests where data fitted the relevant assumptions of normality and homogeneity of variance. Transformations were conducted to achieve normality in some cases, otherwise we used non-parametric tests. For group-level analyses (foraging activity, latrining, group fissions), we ran Wilcoxon signed-rank tests in SPSS 24 (IBM Corp, 2016) and used exact tests to generate $p$-values. When multiple factors and repeated sampling from the same groups and individuals needed to be taken into consideration, we conducted linear mixed models (LMMs) or generalized linear mixed models (GLMMs) [52] in RStudio 3.6.2 (R Core Team, 2019) using the packages lme4 [53] and glmmTMB [54]. See electronic supplementary material for further details about general modelling approach.

To investigate grooming patterns, we initially used separate mixed models for Day 2 and Day 6 of each trial week. We adopted a two-stage process that first entailed analysing the proportion of time that individuals spent grooming using a GLMM with a beta error distribution and either a logit-link function (Day 2) or a probit-link function (Day 6). We then ran further mixed models to investigate whether the increase in proportion of time grooming was due to individuals grooming at a greater rate and/or because bouts were longer. To consider grooming rate per individual, we analysed counts of grooming interactions using GLMMs with a Poisson error distribution and a sqrt-link function, and with log(duration) added as an offset term to account for differences in the time available for grooming. Mean grooming-bout duration per individual was log-transformed and analysed using LMMs with a Gaussian error distribution and an identity-link function. These grooming models all included treatment (Intrusion, Control), dominance status (dominant, subordinate) and sex (female, male), as well as the interactions of treatment with dominance status and sex, as fixed effects; individual identity was nested within group identity as a random effect. Having found intraspecific variation

in how treatment affected grooming on both Day 2 and Day 6 (see Results), we then directly compared grooming on these two days in Intrusion weeks to investigate if the build-up of intergroup threat drove stronger grooming differences. We ran similar models to before (proportion of time grooming: GLMM with a beta error distribution and a logit-link function; grooming rate: GLMM with a Poisson error distribution and a sqrt-link function, with log(duration) as an offset term; log-transformed mean bout duration: LMM with a Gaussian error distribution and an identity-link function). These grooming models all included day (Day 2, Day 6), dominance status and sex, as well as the interactions of day with dominance status and sex, as fixed effects; individual identity was nested within group identity as a random effect.

We used separate mixed models to analyse sentinel and nearest-neighbour data on Day 2 and on Day 6 of each trial week. To analyse whether a sentinel was present when a scan-sample was conducted (yes or no), we used binomial GLMMs with a logit-link function. These models included treatment and controlled for habitat (open, medium, dense), wind (still, light breeze) and group size as fixed effects; group identity was included as a random effect. To analyse individual nearest-neighbour distances from each scan-sample (square-root transformed), we used Gaussian LMMs with an identity-link function. These models included treatment, dominance status and sex, as well as the interactions of treatment with dominance status and sex, as fixed effects; individual identity was nested within group identity as a random effect.

Changes in body-mass gain likely reflect the collective effect of behavioural changes, so we conducted two sets of analyses. First, we analysed individual data on body-mass changes during the morning foraging session for Day 2 and Day 6 separately (as above). We used Gaussian LMMs with an identity-link function, with separate models for adults and for independently foraging pups; raw data were used in all cases, apart from the body-mass change for pups on Day 6 which was square-root transformed. These day-specific body-mass models included treatment and sex, as well as their interaction, as fixed effects; individual identity was nested within group identity as a random effect. For adult models, dominance status and its interaction with treatment were also included as fixed effects. Second, we assessed directly whether there was a difference in response between Day 2 and Day 6. We used separate Gaussian LMMs with an identity-link function for adults and independently foraging pups, to assess the difference in body-mass change between Day 2 and Day 6 of Intrusion and Control weeks. Both these models were run using the raw data and included treatment as a fixed effect; individual identity was nested within group identity as a random effect.

## 3. Results

Grooming behaviour was significantly affected by treatment the day after the first simulated intrusion: a greater proportion of time was spent grooming on Day 2 of Intrusion weeks compared to Control weeks, especially by males (GLMM, treatment × sex: $\chi^2 = 4.030$, $p = 0.045$, estimate ± s.e. = 0.653 ± 0.320; electronic supplementary material, table S1a). This grooming difference was the result of both a greater frequency of grooming bouts (treatment: $\chi^2 = 7.986$, $p = 0.005$, estimate ± s.e. = 0.374 ± 0.131; electronic supplementary material, table S1b) and a longer mean bout duration, especially in males (LMM, treatment × sex interaction: $\chi^2 = 5.270$, $p = 0.022$, estimate ± s.e. = 0.705 ± 0.310; electronic supplementary material, table S1c; figure 2a). Grooming was also significantly affected by treatment on Day 6, when there was still a greater proportion of time invested in

that activity in Intrusion weeks than Control weeks (GLMM, treatment: $\chi^2 = 11.750$, $p = 0.001$, estimate ± s.e. = 0.171 ± 0.049; electronic supplementary material, table S2a). As on Day 2, the treatment-based grooming difference was a consequence of both more frequent grooming (treatment: $\chi^2 = 8.457$, $p = 0.004$, estimate ± s.e. = 0.346 ± 0.118; electronic supplementary material, table S2b) and longer bout durations, especially in males (LMM, treatment × sex: $\chi^2 = 5.501$, $p = 0.019$, estimate ± s.e. = 0.633 ± 0.276; electronic supplementary material table S2c; figure 2b) and subordinates (treatment × status: $\chi^2 = 4.305$, $p = 0.038$, estimate ± s.e. = 0.601 ± 0.298; electronic supplementary material table S2c; figure 2c). In direct comparisons of grooming on Days 2 and 6 of Intrusion weeks, there was no significant effect of day or its interaction with either sex or dominance status (proportion of time grooming: electronic supplementary material, table S3a; grooming rate: electronic supplementary material, table S3b; mean bout duration: electronic supplementary material, table S3c).

On Day 2, there was no significant difference between Control and Intrusion weeks in the amount of group foraging activity (proportion of scan-samples that the group was foraging; Wilcoxon signed-ranks test: $Z = 1.363$, $n = 7$, $p = 0.219$; figure 3a), the amount of sentinel behaviour (whether a sentinel was present when a scan-sample was conducted; GLMM: $\chi^2 = 0.051$, $p = 0.822$, estimate ± s.e. = 0.080 ± 0.355; electronic supplementary material table S4a) and the distance between nearest-neighbours when foraging (LMM: $\chi^2 = 2.345$, $p = 0.126$, estimate ± s.e. = −0.092 ± 0.060; electronic supplementary material table S5a; figure 3b). However, on Day 6, while there was still no treatment difference in the amount of sentinel activity (GLMM: $\chi^2 = 1.891$, $p = 0.169$, estimate ± s.e. = 0.428 ± 0.312; electronic supplementary material table S4b), there was significantly less group foraging activity (Wilcoxon signed-ranks test: $Z = 2.366$, $n = 7$, $p = 0.016$; figure 3a) and individuals foraged significantly closer to other group members (LMM: $\chi^2 = 4.329$, $p = 0.037$, estimate ± s.e. = −0.117 ± 0.056; electronic supplementary material, table S5b; figure 3b) in Intrusion weeks (when there had been prolonged rival-group exposure) compared to Control weeks. The Day 6 treatment difference for nearest-neighbour distances did not differ significantly between dominants and subordinates or between males and females (electronic supplementary material, table S5b).

Groups spent a significantly greater proportion of time latrining during Intrusion weeks (median = 0.015, IQR = 0.009) compared to Control weeks (median = 0.004, IQR = 0.004; Wilcoxon signed-ranks test: $Z = 2.028$, $n = 7$, $p = 0.047$). This was the result of a greater rate of latrining ($Z = 2.197$, $n = 7$, $p = 0.031$), not an increase in the mean latrine duration ($Z = 0.676$, $n = 7$, $p = 0.578$). Group fissioning rate was greater in Intrusion weeks (median = 0.002, IQR = 0.001) compared to Control weeks (median = 0.001, IQR = 0.001), but the result was not statistically significant ($Z = 1.859$, $n = 7$, $p = 0.078$).

On Day 2, the body-mass change for adults (LMM: $\chi^2 = 0.022$, $p = 0.883$, estimate ± s.e. = 0.140 ± 0.801; electronic supplementary material, table S6a; figure 4a) and independently foraging pups ($\chi^2 = 0.008$, $p = 0.930$, estimate ± s.e. = −0.086 ± 1.167; electronic supplementary material, table S6c; figure 4b) did not differ significantly between Intrusion and Control weeks. However, on Day 6, both adults ($\chi^2 = 4.198$, $p = 0.040$, estimate ± s.e. = −1.742 ± 0.835; electronic supplementary material, table S6b; figure 4a) and independently foraging pups ($\chi^2 = 4.876$, $p = 0.027$, estimate ±

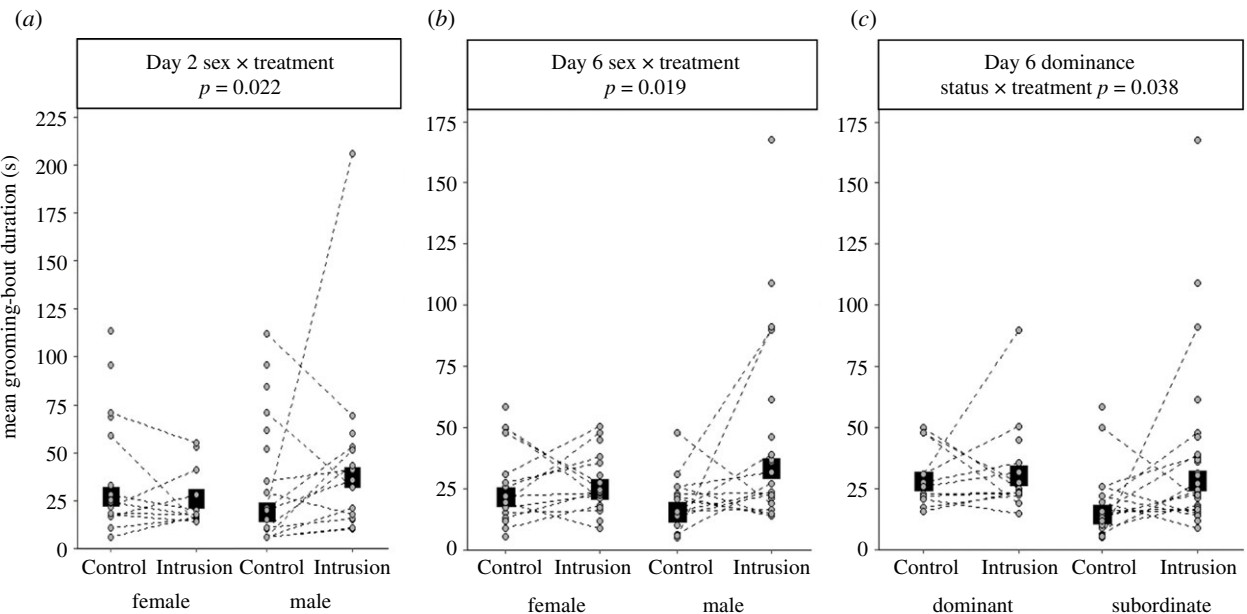

**Figure 2.** Effect of treatment (Control: herbivore; Intrusion: rival group) on dwarf mongoose grooming behaviour. (a) On Day 2 of Intrusion weeks (the day after the first simulated intrusion), grooming bouts were longer than those in Control weeks, but the effect was more pronounced for males than females. On Day 6 of Intrusion weeks (after repeated simulated intrusions), the grooming-bout duration was longer than in Control weeks, but the effect was more pronounced both (b) for males than females and (c) for subordinates than dominants. Shown are back-transformed predicted means (square points) ± s.e. (within the bounds of the square point in some cases) from the mixed models presented in electronic supplementary material, table S1 (for a) and S2 (for b,c), with the raw data (circular points) for each individual. Dashed lines connect data from the same individuals; orphan points are instances where data were available from an individual in only one treatment. In (a), n = 67 mean bout durations from 44 individuals in seven groups; in (b,c), n = 73 mean bout durations from 47 individuals in seven groups.

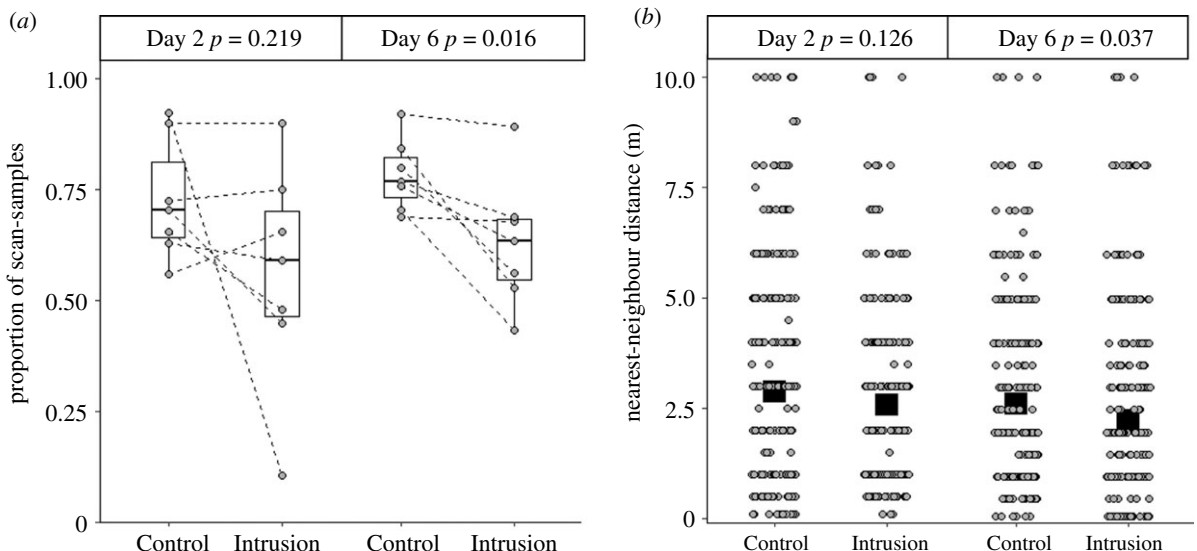

**Figure 3.** Effect of treatment (Control: herbivore; Intrusion: rival group) on dwarf mongoose foraging behaviour. On Day 6 of Intrusion weeks (after repeated simulated intrusions), there was (a) less foraging activity (proportion of scan-samples where the group was recorded as foraging), and (b) individuals foraged closer to one another (nearest-neighbour distance) compared to Control weeks; these effects were not apparent on Day 2. Shown in (a) are values for each group (circular points; n = 7), with dashed lines connecting data from the same groups; boxplots indicate the median and quartiles; whiskers represent data within quartiles ± 1.5 times the interquartile range. Shown in (b) are back-transformed predicted means (square points) ± s.e. (within the bounds of the square point) from the mixed models presented in electronic supplementary material, table S4, alongside raw data (circular points; jitter function applied to spread the points horizontally). n = 451 nearest-neighbour distances from 53 individuals in seven groups on Day 2 and n = 490 nearest-neighbour distances from 54 individuals in seven groups on Day 6.

s.e. = −0.566 ± 0.254; electronic supplementary material, table S6d; figure 4b) gained significantly less body mass in Intrusion weeks compared to Control weeks. There was no significant difference between adult individuals of different dominance status and sex in this Day 6 treatment effect (electronic supplementary material, table S6b). When considering the difference between Day 2 and Day 6 directly, there was a significant reduction in body-mass gain in Intrusion weeks compared to Control weeks for both adults ($\chi^2 = 5.728$, $p = 0.017$, estimate ± s.e. = −2.930 ± 1.199; electronic supplementary material, table S7a) and independently foraging

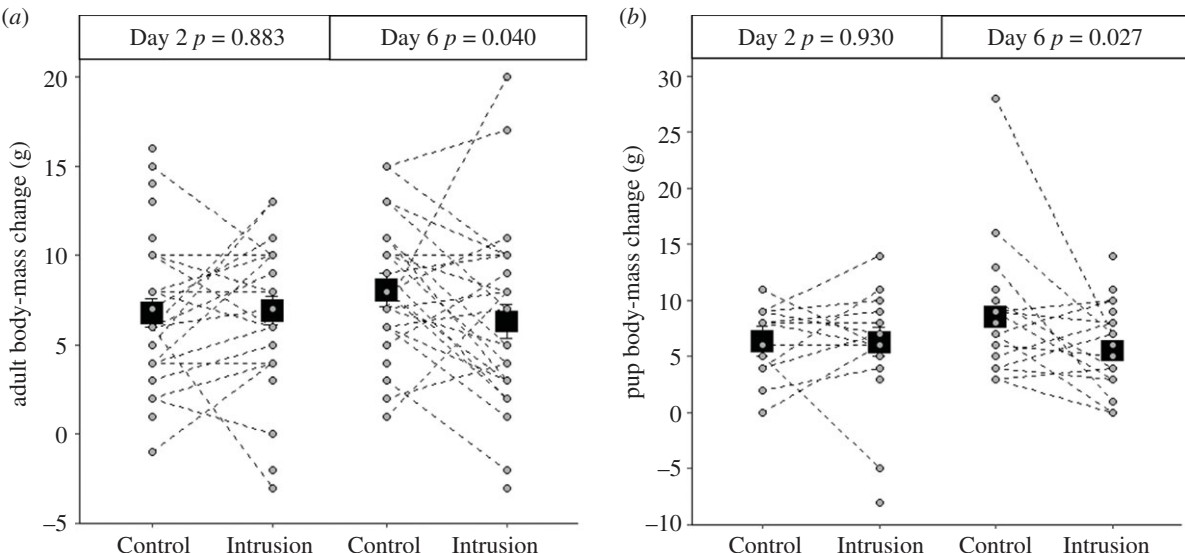

**Figure 4.** Effect of treatment (Control: herbivore; Intrusion: rival group) on dwarf mongoose body-mass changes. On Day 6 of Intrusion weeks (after repeated simulated intrusions), (*a*) adults and (*b*) independently foraging pups gained less body mass compared to Control weeks; these effects were not apparent on Day 2. Shown are predicted means (square points) ± s.e. (within the bounds of the square point in some cases) from the mixed models presented in electronic supplementary material, table S5 (back-transformed for pups on Day 6), with the raw data (circular points) for each individual. Dashed lines connect data from the same individuals; orphan points are instances where data were available from an individual in only one treatment. In (*a*), $n = 62$ body-mass changes from 39 individuals in seven groups on Day 2 and $n = 68$ body-mass changes from 39 individuals in seven groups on Day 6. In (*b*), $n = 38$ body-mass changes from 25 individuals in seven groups on Day 2 and $n = 43$ body-mass changes from 26 individuals in seven groups on Day 6.

pups ($\chi^2 = 7.090$, $p = 0.008$, estimate ± s.e. = $-3.861 \pm 1.333$; electronic supplementary material, table S7b).

## 4. Discussion

We provide strong experimental evidence for extended and cumulative effects of intergroup conflict on within-group behaviour in dwarf mongooses. By looking beyond the immediate aftermath of outgroup interactions, we expand on previous research looking at the short-term behavioural consequences in two main ways. First, we demonstrate that exposure to a single simulated rival-group intrusion can have an extended effect into the following day: compared to their behaviour in Control weeks, dwarf mongooses invested more in grooming their groupmates on Day 2 of Intrusion weeks. Second, we show that exposure to repeated rival-group intrusions can lead to further differences in baseline behaviour: by Day 6, there were treatment differences in foraging—groups exhibited reduced foraging activity and group members foraged in closer proximity to one another in Intrusion weeks compared to Control weeks—that were not apparent on Day 2. In addition, compared to Control weeks, groups spent more time in Intrusion weeks scent-marking their territories with a resulting tendency to fission more often. Consequently, individuals (both adults and independently foraging pups) gained less body mass on Day 6 in Intrusion weeks than in Control weeks; there was a greater change in this body-mass effect between Day 2 and Day 6 in Intrusion weeks compared to Control weeks. Together, these results indicate a cumulative effect of intergroup conflict.

There are potential proximate and ultimate explanations for the behavioural changes seen. Increased affiliation could be driven proximally by conflict-induced chronic stress, as giving and receiving grooming can reduce anxiety [27,29],

or functionally because increased affiliation could incentivize future help in signalling or adversarial interactions with outsiders [10,11,30]. Individuals may also choose to forage in closer proximity to groupmates if that leads to a reduction in conflict-induced anxiety [55,56] or enhances the likelihood of support were an intergroup contest to arise [12]. Increasing advertisement of territory ownership through latrining behaviour may help to establish group dominance and thus increase the likelihood of resource retention [57,58]. The reduction in foraging activity probably reflects trade-offs with other behaviours, such as moving, latrining and grooming, which ultimately reduce the time available for foraging [19,59] and could have caused the reduction in body-mass gain. A sustained decline in body-mass gain during periods of intense intergroup pressure could have fitness consequences, increasing the vulnerability of individuals to predation and disease [60]. In addition, since many cooperative behaviours, such as babysitting, sentinel duty, offspring feeding and territorial defence, are state dependent [61–63], individuals in poorer body condition might invest less in these activities with negative consequences for groupmates and their own inclusive fitness.

The increased grooming on Day 2 of Intrusion weeks was in line with our prediction that there would be some lasting changes to individual behaviour after a single simulated intrusion. One explanation for increased grooming in the immediate aftermath of intergroup conflict is that GC levels are elevated [27,29], but this is unlikely to explain the extended effect that we documented. After a stressful stimulus, it is common for GCs to return from a peak to baseline within a couple of hours, although the more stressful the stimulus the longer it takes [64]. A single intergroup faecal presentation is unlikely to induce an intense stress response, and our presentation was in the evening of Day 1 with the grooming data collected during Day 2. Moreover, if the extended effect on grooming was due to a sustained GC

response, we would expect to see other behavioural changes on Day 2 but we found no increase in sentinel activity or a reduction in nearest-neighbour foraging distance; results which were apparent in the hour after a simulated intergroup threat, when GCs are likely to be elevated [12]. Instead, there could be a functional explanation for the increase in grooming on Day 2. While delayed rewarding is known to occur in this species [49], rewarding is unlikely to be relevant here as there was no intergroup contest that required involvement on Day 1. Rather, the increased grooming could be pre-emptive affiliation to increase groupmate participation in future contests [30,34], especially if the likelihood of a confrontation arising is greater having recently discovered fresh cues to the presence of a rival group. Data from vervet monkeys (Chlorocebus pygerythrus) show that females can influence subsequent male participation during extended intergroup encounters by grooming them [11], but further work is needed to determine whether grooming can be used to promote participation in future contests that occur hours or days later.

Also as predicted, we found that repeated exposure to intergroup threats over the course of a week resulted in further changes. There were treatment differences in foraging and body-mass gains on Day 6 that were not apparent on Day 2, and collective latrining was also greater during Intrusion weeks compared to Control weeks. The behavioural changes occurred outside the immediate post-conflict period (i.e. the period following interaction with rival-group cues), meaning that baseline behaviour was affected by intergroup threat. A possible proximate explanation relates to the influence of elevated GCs. Unlike on Day 2, where any stimulus-induced increase in GCs from the previous evening should have subsided (see above), the repeated discovery of cues from a rival group could lead to a build-up in GCs and chronic stress [32,65]. Foraging closer to other individuals could help to reduce anxiety; the reduction in body-mass gain could also be a direct effect of elevated GCs [65], alongside an indirect effect of greater investment in non-foraging related activities (e.g. latrining). From a functional perspective, the documented behavioural changes provide evidence for increased cooperation during periods of heightened intergroup conflict [66–68]. These experimental findings which are in-line with previous observational studies reporting a positive correlation between outgroup interactions and various measures of in-group cohesion [14,34], along with empirical work focussing on the immediate post-contest period [13,15,16].

We found evidence for intragroup variation in grooming responses to intergroup threats on both Day 2 and Day 6. Asymmetries in affiliative behaviour depending on individual characteristics have been reported in other nonhuman species both during and in the immediate aftermath of outgroup interactions [11,14,16]. However, prior work on this study population did not find any differences in grooming based on dominance status or sex in the hour after a single simulated intergroup threat [12]. Instead, differences in dwarf mongooses seem to emerge later; on Day 6, those differences are probably because the perceived threat to particular individuals is greater after repeated simulated intrusions, but there was also already a sex difference in grooming response on Day 2. Against expectations, the increase in grooming behaviour was greater for males compared to females and for subordinates compared to dominants. Although we do not know how participation in intergroup contests differs among dwarf mongooses, males of many species contribute more than females [1,7] and, for cooperatively breeding species, subordinate helpers often participate more than dominant group members [2,69]. Therefore, these individuals could be pre-emptively preparing for an upcoming contest. Consolidating social bonds prior to a contest could promote participation during the encounter, which could in turn increase the likelihood of winning [14,15]. It has also been shown that going into battle alongside established partners mediates the stress response, resulting in lower GCs afterwards than if participating without bond partners [27]. Our individual-based results add to a growing body of evidence that demonstrates how group heterogeneity leads to diverging behavioural consequences when exposed to intergroup threats [70] and highlights the importance of considering intragroup variation in the study of outgroup conflict.

Overall, our results suggest that intergroup conflict can have longer-lasting behavioural consequences than previously documented, either through extended effects from single events or as a result of the cumulative build-up of threat. While it is possible that even stronger responses might have been found if focal groups had encountered rival mongooses, the potential presence of rivals (as indicated by faecal presentations and call playbacks) was sufficient to generate at least some behavioural and body-mass changes. This also showcases the value of such experimental methods, which are feasible in natural conditions and do not require the potentially stressful process of translocating individuals for live intrusions. Future experimental studies need to move beyond behavioural responses and measure fitness consequences directly; to investigate how cumulative outgroups threats can impair, for example, immune function and growth [33] and, ultimately, impact survival and reproductive success [3,71].

Ethics. The study was undertaken by permission from the Department of Environmental Affairs and Tourism, Limpopo Province (permit no. 001-CPM403-00013) and the Ethical Review Group, University of Bristol (University Investigator no. UIN/17/074).

Data accessibility. Data are available from the Dryad Digital Repository: https://doi.org/10.5061/dryad.98sf7m0k4 [72] and in the electronic supplementary material [73].

Authors' contributions. A.M.-D.: conceptualization, data curation, formal analysis, investigation, methodology, project administration, validation, visualization, writing—original draft, writing—review and editing; J.F.L.: investigation, project administration, writing—review and editing; J.M.K.: project administration, supervision, writing—review and editing; A.N.R.: conceptualization, funding acquisition, methodology, project administration, resources, supervision, validation, writing—review and editing.

All authors gave final approval for publication and agreed to be held accountable for the work performed therein.

Competing interests. We declare we have no competing interests.

Funding. This work was funded by a European Research Council Consolidator Grant (no. 682253) awarded to A.N.R.

Acknowledgements. We thank B. Rouwhorst and H. Yeates for access to their land, J. Arbon, F. Bayliss, A. Byrne, E. Richens and H. Rickets for invaluable support and assistance in the field, M. Aveling for the beautiful figure illustrations, and I. Braga-Goncalves for useful comments on the manuscript.

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
