## [Peer Review File · Proceedings of the Royal Society B: Biological Sciences]

Review History

RSPB-2021-1743.R0 (Original submission)

Review form: Reviewer 1

Recommendation

Major revision is needed (please make suggestions in comments)

Scientific importance: Is the manuscript an original and important contribution to its field?

Excellent

General interest: Is the paper of sufficient general interest?

Excellent

Quality of the paper: Is the overall quality of the paper suitable?

Good

Is the length of the paper justified?

Yes

Should the paper be seen by a specialist statistical reviewer?

No

Do you have any concerns about statistical analyses in this paper? If so, please specify them explicitly in your report.

Yes

It is a condition of publication that authors make their supporting data, code and materials available - either as supplementary material or hosted in an external repository. Please rate, if applicable, the supporting data on the following criteria.

Is it accessible?

Yes

Is it clear?

Yes

Is it adequate?

Yes

Do you have any ethical concerns with this paper?

No

Comments to the Author

Comments on RSPB-2021-1743

This study investigates the effects of simulated intergroup conflict on within-group affiliative and social behaviour using field experiments in a population of wild dwarf mongooses (*Helogale parvula*). The authors test whether exposure to a single rival group stimulus leads to longer-term changes in grooming, foraging and vigilance behaviour, and weight gain (over the course of the next day), and whether repeated exposure to rival group stimuli results in a cumulative effect on within-group behaviour. They find evidence for both 'carryover' and cumulative effects in groups and, interestingly, that behavioural responses are different for different individuals (dominants versus subordinates, and males versus females). They conclude that intergroup threat has the potential to serve as a strong selective force shaping within-group behaviour and fitness.

I congratulate the authors on designing and performing a neat experiment with appropriate and well thought out controls – a great effort in the field where so much can go wrong, particularly when conducting on a week-long trial. While sample sizes are not huge (particularly for the group-level metrics), the value of these types of experimental studies are notable and the results can provide an important and interesting contribution to the field when considered carefully. I found the paper was well written and enjoyable to read. The part where I felt the paper fell down was in the analysis of the data, particularly relating to testing cumulative effects, and in the statistical models and approach used to test hypotheses. I feel there are some areas to address in this regard before it can be considered for publication, and I outline these here:

Main comments:

1. The test of cumulative effects of repeated exposure to simulated threat on day 6 does not directly test for cumulative effects because there is no direct statistical comparison between day 2 behaviour and day 6 behaviour (only day 6 Intrusion trials to day 6 Control trials). This makes any conclusions about there being a cumulative effect difficult to make, and therefore statements of there being a "stronger treatment difference on day 6 than day 2" (e.g. L329-331, L347-349) are unsupported. Currently, the results show that mongooses respond to repeated exposures to threats (as compared to repeated exposure to a benign stimulus), but not that this response is stronger than after just one exposure (as is claimed by a observed cumulative effect). I believe it is possible to test for a cumulative threat by analysing day 2 and day 6 data from Intrusion trials in one analysis (for each response variable), and including time as the explanatory variable of interest (perhaps also including interactions between time and dominance status or sex to investigate inter-individual responses as per the original analyses). Alternatively, day 2 and day 6

data for both Intrusion and Control trials could be included in one analysis (for each response variable) and the interaction between time and treatment used to test whether the behavioural response on day 6 is different to day 2 (for Intrusion trials, as predicted, but not for Control trials). The potential problem with this approach is that testing for inter-individual differences would involve testing 3-way interactions (e.g. time*treatment*sex) and 3-way interactions are complicated and tricky to interpret. My recommendation would be to include an analysis in addition to those already performed that explicitly tests data from Intrusion trials for day 2 and day 6, as I described. This would mean that claims of a cumulative effect can be supported statistically, rather than inferred from two separate analyses (and plots of data).

2. The types of models (transformations of data in a model with a Gaussian error structure) used to analyse the proportion of time grooming and the rate of grooming in linear models is not the most appropriate given the distribution of the data. Neither of these variables are normally distributed by definition (proportion time is bounded by 0 and 1, and a rate is a count of occurrences per unit time). A more appropriate model to use for the proportion time data is one with a beta error distribution which accommodates bounded variables (e.g. Smithson & Verkuilen 2006 <https://doi.org/10.1037/1082-989X.11.1.54>). Similarly, for the rate of grooming, a more appropriate model is one with a Poisson error structure that includes an offset (use 'offset' in R) of the exposure time (the time available to groom, log transformed) as an additional fixed effect. Using an offset in a Poisson model is a conventional way of analysing rate data (See Crawley MJ. 2007. *The R book*, Chapter 9 of Zuur et al. 2009. *Mixed effects models and extensions in ecology with R*, and numerous online forums).

3. I have concerns with the stepwise elimination of fixed effects conducted as part of statistical analyses to obtain a minimal model. The use of stepwise elimination of non-significant terms (specifically main effects) is widely accepted that this method of model simplification is flawed because it often results in overestimated effect sizes, multiple hypothesis testing and inflated type I error rates. See these papers for discussion:

Forstmeier, W., & Schielzeth, H. (2011). Cryptic multiple hypotheses testing in linear models: Overestimated effect sizes and the winner's curse. *Behavioral Ecology and Sociobiology*, 65(1), 47e55. <http://dx.doi.org/10.1007/s00265-010-1038-5>.

Mundry, R., & Nunn, C. L. (2009). Stepwise model fitting and statistical inference: Turning noise into signal pollution. *American Naturalist*, 173(1), 119e123. <http://dx.doi.org/10.1086/593303>.

Whittingham, M. J., Stephens, P. A., Bradbury, R. B., & Freckleton, R. P. (2006). *Journal of Animal Ecology*, 75(5), 1182e1189.

While removing non-significant interaction terms in order to test main effects is accepted (see Engquist (2005) *Animal Behaviour* 70, 967-971 doi:10.1016/j.anbehav.2005.01.016), removing non-significant main effects changes the partitioning of variance between the remaining main effects and so can lead to the problems I have outlined above. My recommendation would be to retain non-significant main effects in the model and report the estimates, standard errors, test statistics and p-values from these full models.

Other comments:

4. Title and throughout manuscript: I am not sure that the use of the term 'carryover' to capture the effects observed the day after exposure to a rival threat is the most appropriate. Carryover effects in ecology are usually used for effects observed in one breeding season or part of the life history that have been influenced by the organism's experience in the breeding season (or part of life history) before. Would 'longer term' or 'delayed' effects in the context of this study be more appropriate?

5. L45: see also Preston et al. 2020 *Journal of Animal Ecology*, 90 153-167 DOI: 10.1111/1365-2656.13323 for a study that examines longer term consequences of simulated intergroup conflict on grooming behaviour.

6. L119-126: Can you give directional predictions alongside the rationale, rather than just saying that behavioural responses will be 'greater' following multiple exposures, or for certain

individuals. Also, it would be nice to include predictions of the effect on rarer group latrining and fissioning events. At the moment these seem like a bit of an afterthought because these behaviours were able to be observed rather than forming part of the set up of the study.

7. L142: I'm not sure I understand the statement that the playback was not used for data collection purposes. Why use the playback at all?
8. L153-154: What is the rationale for using a non-neighbouring group as the rival threat, rather than a rival group that shared a territorial boundary with the focal group? Does a non-neighbour represent a larger threat (i.e. a dear enemy effect)?
9. L162: 'a playback' or 'playbacks'.
10. L175: Can you provide mean duration (and range) between trials?
11. L187-190: Can you provide means +/- SE for the treatments in the results for Wilcoxon tests so that the reader can more easily evaluate the strength of the effect.
12. L201-203: How confident can you be that the observer was able to record all interactions and behaviours throughout the day as (potentially) the only observer while the group was foraging and potentially moving in and out of sight?
13. L226-227: Can you provide more information on how P-values were generated using the Monte Carlo resampling method? This seems like an important part of analyses and subsequent interpretation of data (P-values are used to determine significance) and so more detail is needed on how this was performed.
14. L230-231: More information is required on the models in the main text of the manuscript, rather than relying on this information being accessed in the ESM. The minimum amount of information provided on statistical models should be the response variable, the type of model (e.g. the error structure used), and the fixed and random effects. Without this information it is difficult to evaluate the results and the subsequent interpretation. To me, this information is critical to the paper and should be provided to the reader in the main text. In general I found the statistical methods not particularly well explained and quite difficult to follow because the information is scattered throughout the main text, ESM and ESM table headings.
15. L236 and throughout: Include model estimates and SEs with statistical results so that the reader can more easily evaluate the strength of the effect. The test statistic and P-value alone is not sufficient to assess effect size which is arguably a more informative metric than the P-value.
16. L242 and throughout: Use 'status' rather than 'dominance' when referring to the dominance status variable to be consistent with terminology in the ESM.
17. L242: The stats report here are different to the ones in Table S2a in the ESM.
18. L250: I think you mean the probability of the group foraging at each scan sample.? As I understand, this is what you tested in your binomial model; the proportion of scans that the group was observed foraging is a subtly different metric (as is the amount of foraging).
19. L251: As above, do you mean the probability of the group sentinelling at each scan?
20. L249-252 & L253-256: I am very surprised that you have been able to run a GLMM with only 14 data points for these analyses, particularly with a number of fixed and random effects. Additionally, you don't provide any estimates for the effects that you are reporting here, even in the ESM (perhaps because you remove them from the model?). This is important information to report - non-significant effects will have an estimate from the model and if you are reporting, and particularly plotting, these results then this information should be included.
21. L329-331 & L347-349: I am not convinced that you have shown that the effect on day 6 is stronger than that on day 2 (see my main comment 1 above).
22. L356-360: Do you know with whom males/subs are grooming? Can you comment on whether there might be some consolidating of bonds between, e.g. subs and subs who might be more involved in an intergroup interaction?
23. L366-367: Might an appropriate reference here be Thompson & Cant 2018 Behav Ecol doi:10.1093/beheco/ary044
24. L369-371: What is the frequency of naturally occurring intergroup interactions in this species, and how relevant is it to experience a barrage of threat stimuli in a short space of time? How might the response of dwarf mongooses to repeated stimuli compare to other species in

which intergroup fighting takes place very often, and is more intensely aggressive (e.g. banded mongooses, meerkats)? Perhaps the strength of the response in dwarf mongooses might be larger because rivals are not encountered as frequently? Including some information on the frequency of intergroup interactions somewhere in the manuscript (possibly when describing the system) might help to further contextualise the results.

25. All figures: Could you also plot the predicted means from your models as well as the raw data. Currently there are a number of plots where there is a statistically significant difference between treatments, but the difference between the raw data does not look very convincing (e.g. Fig 1b - there does not appear to be much difference in the relationships of doms and subs across control and intrusion trials; Fig 2c day 6 panel). Plotting the estimates (with standard error) from your models, alongside raw data, might allow statistically significant differences between categories to be more apparent because the estimated mean will be different from the median of the boxplots that are currently presented.

26. Fig 1: If an individual only groomed in one treatment then they will have a matching point at zero for the treatment in which they did not groom (i.e. there won't be any orphan points; data points at zero are still data points and should not be removed). Do you mean that some individuals were only observed in one trial and therefore only contributed one data point?

27. Fig 2a: y-axis should be probability of group foraging rather than proportion of group foraging activity? Similarly for Fig 2b, probability not proportion?

28. Fig 4: I am not able to marry these figures up with descriptions of the analysis because there is little (no) description of the way in which these rare events were analysed, either in the main text or the ESM.

29. ESM: In the Data Analysis section, I would like it explained explicitly what fixed and random effects were included in each model with some rationale as to why. Currently, I am unclear as to what variables were included in analyses and have to refer to ESM tables to find out. I think this information is better presented in the main text (see my comment above).

30. ESM: In the data analysis section, you describe sequentially removing terms using drop1 but drop1 does not remove the terms, it tests the effect of removing each fixed effect in turn against the full model containing that variable. This is what you describe as the 'ANOVA model comparison', but at the moment it sounds like drop1 is used to remove effects (rather than decide which effects to remove). I have my objections to stepwise elimination of variables to simplify models, which I have already outlined.

31. It is not clear why you refer to using Akaike Information Criteria to evaluate fixed effects when there is no reporting of any AIC values for any models. Throughout the study, P-values are used to determine the significance of effects and so to also use AIC values conflates two very distinct statistical approaches and philosophies. Can you remove reference to AIC values to avoid confusion?

32. ESM: I do not understand this sentence: "We added non-significant effects individually back to the minimal model to obtain significance levels, whilst we obtained values for significant effects either by comparing the term with a null model or by comparing the full minimal model with each term removed individually." What do you mean by 'values for significant effects'?

33. ESM: Why do you round nearest neighbour distances that are less than 0.5m to 0.1m rather than rounding to 0?

34. ESM: When calculating body mass change, might you have a problem with regression to the mean whereby extremely high or low values in the first measure of a given variable are more likely to move closer to the mean in a second measure of that variable. You can control for regression to the mean (e.g. Kelly & Price 2005 *Am Nat* 166, 700-707 doi:10.1086/497402).

35. ESM Tables: you should give details of any data transformations in the methods (ideally of the main text) rather than burying this information in an ESM table heading.

36. ESM Tables: What are your reference levels for sex and dominance status?

37. ESM Tables: Please report sample size with statistical tables.

38. ESM Tables: Why are effect sizes and standard errors for removed variables missing?

This is important information, particularly when these patterns are reported and plotted in the manuscript, and so needs to be included.

Review form: Reviewer 2

Recommendation

Major revision is needed (please make suggestions in comments)

Scientific importance: Is the manuscript an original and important contribution to its field?

Excellent

General interest: Is the paper of sufficient general interest?

Excellent

Quality of the paper: Is the overall quality of the paper suitable?

Good

Is the length of the paper justified?

Yes

Should the paper be seen by a specialist statistical reviewer?

No

Do you have any concerns about statistical analyses in this paper? If so, please specify them explicitly in your report.

Yes

It is a condition of publication that authors make their supporting data, code and materials available - either as supplementary material or hosted in an external repository. Please rate, if applicable, the supporting data on the following criteria.

Is it accessible?

Yes

Is it clear?

Yes

Is it adequate?

Yes

Do you have any ethical concerns with this paper?

No

Comments to the Author

This manuscript is another fascinating and well thought through and implemented experiment from the Radford group that can be used as a showcase of how to conduct experiments. Here Amy Morris-Drake and colleagues investigate the behavioural long-term effects of inter-group conflict using an excellent paradigm to investigate whether or not this effect is different after a single conflict or after the accumulation of several conflicts. They find several behavioural differences, some of which change due to a likely cumulative effect. The data and the results are convincing, although some questions remain regarding to the statistical analysis, which to my understanding need some improvement. Their conclusions are driven by the results and explained in easy understandable and well structured sentences. Overall, this paper is a very good example on how such a topic can be researched and such questions can be addressed by an experiment.

I have three major points with regards to the statistical analysis:

(1) with small data sets like this (N=7) exact tests should be conducted. This is important because

distributions with such a small sample size are not easy to assess. Please check Mundry & Fischer 1998 in *Animal Behaviour*

(2) the authors seem to undergo a model selection process removing non-significant fixed effects. While this is usual procedure for non significant interactions of fixed effects, this is seen as multiple testing when done with fixed effects, whether they are test or control predictors. I suggest to follow a similar procedure like in Wedgell et al. 2019 for an experiment in Guinea baboons.

(3) an eloquent way of checking whether or not your predictors of interest are working well is to test explanatory power in a full-null model comparison. The authors have some clear predictions and should therefore do so. This makes a complex model very simple reduced to one test. Here the authors could also follow the methods described in Wedgell et al. 2019

I believe that the results are very robust and most of the results will stand with this statistical method as well. The advantage of this method is that the results will not be seen as stronger and more rigorous.

I have to state clearly that this paper is truly an advance in the study of inter-group conflict and that I really enjoyed reading and thinking about it. I fully support its publication.

Specific comments:

line 86-87: I am not entirely sure what the argument here is: it is true that a cumulative effect is the result of shorter latency of conflicts - but by definition this is also what an experimental approach is doing - it is increasing the frequency of inter-group conflict by adding additional conflicts to the naturally occurring ones. The advantage of the experiment on the other side is that the amount and the numbers can be obviously controlled - see for example Wilson et al 2001, Herbinger et al. 2009

line 129 - 149: A figure of the experimental protocol would be good, since it is a bit confusing how often and for which duration the test (intrusion) and the control stimuli have been presented: (in the beginning it is on day 1-5 of the week, over two weeks, than in the control it is suddenly only one week - talking about the control week, and later (line 140) there is a range of 3-5 days of feacal presentation. In addition there seems to be a variation of playback experiments that have been conducted per group, potentially creating a different type of threat (immediate) compared to the feacale presentation.

line 194: DMRP = ?

line 347 - 350: It would be interesting to see if the groups had changed their ranging behaviour as well. I assume that the burrow of mangose is more or less in the middle of their territory and having signs of a stranger group at the burrow, in the center of the territory, must be more frightening than in the periphery. So after a week of reoccurring sign of competitors at the own burrow - are they more likely to stay close to the burrow, to not loose it suddenly to unwanted guests?

Decision letter (RSPB-2021-1743.R0)

14-Sep-2021

Dear Miss Morris-Drake:

Your manuscript has now been peer reviewed and the reviews have been assessed by an Associate Editor. The reviewers' comments (not including confidential comments to the Editor) and the comments from the Associate Editor are included at the end of this email for your

reference. As you will see, the reviewers and the Editors have raised some concerns with your manuscript and we would like to invite you to revise your manuscript to address them.

Research ethics:

Use of animals and field studies:

It is a condition of publication that you make available the data and research materials supporting the results in the article. Please see our Data Sharing Policies (<https://royalsociety.org/journals/authors/author-guidelines/#data>). Datasets should be deposited in an appropriate publicly available repository and details of the associated accession number, link or DOI to the datasets must be included in the Data Accessibility section of the article (<https://royalsociety.org/journals/ethics-policies/data-sharing-mining/>). Reference(s) to datasets should also be included in the reference list of the article with DOIs (where available).

Please submit a copy of your revised paper within three weeks. If we do not hear from you within this time your manuscript will be rejected. If you are unable to meet this deadline please let us know as soon as possible, as we may be able to grant a short extension.

Best wishes,
Dr Locke Rowe
mailto:proceedingsb@royalsociety.org

Associate Editor
Board Member: 1
Comments to Author:

I have now received thorough reviews from two experts. Both reviewers were positive about the topic and complimentary on the methods and quality of the manuscript. The concerns raised were associated with the statistical analyses and additional clarification of methods. From my own reading of the manuscript, I am in agreement with the reviewer comments and concerns.

Reviewer 2 raises an important point about inferring differences between time periods based on analyses within time periods, rather than directly assessing the differences. Their suggestions are reasonable and should allow you to directly test whether there are indeed differences between day 2 and day 6. This is necessary for any comparison among times.

Both reviewers commented on the potential issues associated with the removal of nonsignificant fixed effects. I am in agreement here that these fixed effects should not be removed (although nonsignificant interactions with them can be). In addition to the statistical concerns raised, these fixed effects are presumably biologically relevant to the study (thus their initial inclusion) and were part of the original design. Their inclusion also allows for comparison of their influence across behaviors and times in a more standard way.

Reviewer 1 suggests using different methods potentially better suited to the grooming data and these are well-worth exploring. If patterns are consistent across methods and the transformations resulted in well-behaved residuals in the original analyses, I do not feel strongly about needed to replace the analyses. However, it would be helpful to include these additional methods in the ESM if you decide to retain the original analyses.

Reviewer 1 suggests a number of areas in the methods and results that need further clarification or information. I am also in agreement with reviewer 2 that a figure illustrating the experimental design would be very helpful – perhaps a smaller and less detailed version of Figure S1 in the ESM? Additionally, information defining ‘close’ and ‘lost’ calls from the ESM should be included in the main text (line 162) as well.

I think this is a solid, novel study with great appeal to a broad audience. I encourage the authors to pay careful attention to the issues raised above, as well as the other helpful comments so thoughtfully provided by the reviewers. I look forward to seeing a revised version of this study soon.

Sincerely,
Katie McGhee
Associate Editor

Reviewer(s)¹ Comments to Author:
Referee: 1

Comments to the Author(s)
Comments on RSPB-2021-1743

This study investigates the effects of simulated intergroup conflict on within-group affiliative and social behaviour using field experiments in a population of wild dwarf mongooses (*Helogale parvula*). The authors test whether exposure to a single rival group stimulus leads to longer-term changes in grooming, foraging and vigilance behaviour, and weight gain (over the course of the next day), and whether repeated exposure to rival group stimuli results in a cumulative effect on within-group behaviour. They find evidence for both ‘carryover’ and cumulative effects in groups and, interestingly, that behavioural responses are different for different individuals (dominants versus subordinates, and males versus females). They conclude that intergroup threat has the potential to serve as a strong selective force shaping within-group behaviour and fitness.

I congratulate the authors on designing and performing a neat experiment with appropriate and well thought out controls – a great effort in the field where so much can go wrong, particularly when conducting on a week-long trial. While sample sizes are not huge (particularly for the group-level metrics), the value of these types of experimental studies are notable and the results can provide an important and interesting contribution to the field when considered carefully. I found the paper was well written and enjoyable to read. The part where I felt the paper fell down was in the analysis of the data, particularly relating to testing cumulative effects, and in the statistical models and approach used to test hypotheses. I feel there are some areas to address in this regard before it can be considered for publication, and I outline these here:

Main comments:

1. The test of cumulative effects of repeated exposure to simulated threat on day 6 does not directly test for cumulative effects because there is no direct statistical comparison between day 2 behaviour and day 6 behaviour (only day 6 Intrusion trials to day 6 Control trials). This makes any conclusions about there being a cumulative effect difficult to make, and therefore statements of there being a “stronger treatment difference on day 6 than day 2” (e.g. L329-331, L347-349) are unsupported. Currently, the results show that mongooses respond to repeated exposures to threats (as compared to repeated exposure to a benign stimulus), but not that this response is stronger than after just one exposure (as is claimed by a observed cumulative effect). I believe it is possible to test for a cumulative threat by analysing day 2 and day 6 data from Intrusion trials in one analysis (for each response variable), and including time as the explanatory variable of interest (perhaps also including interactions between time and dominance status or sex to investigate inter-individual responses as per the original analyses). Alternatively, day 2 and day 6 data for both Intrusion and Control trials could be included in one analysis (for each response variable) and the interaction between time and treatment used to test whether the behavioural response on day 6 is different to day 2 (for Intrusion trials, as predicted, but not for Control trials). The potential problem with this approach is that testing for inter-individual differences

would involve testing 3-way interactions (e.g. time*treatment*sex) and 3-way interactions are complicated and tricky to interpret. My recommendation would be to include an analysis in addition to those already performed that explicitly tests data from Intrusion trials for day 2 and day 6, as I described. This would mean that claims of a cumulative effect can be supported statistically, rather than inferred from two separate analyses (and plots of data).

2. The types of models (transformations of data in a model with a Gaussian error structure) used to analyse the proportion of time grooming and the rate of grooming in linear models is not the most appropriate given the distribution of the data. Neither of these variables are normally distributed by definition (proportion time is bounded by 0 and 1, and a rate is a count of occurrences per unit time). A more appropriate model to use for the proportion time data is one with a beta error distribution which accommodates bounded variables (e.g. Smithson & Verkuilen 2006 <https://doi.org/10.1037/1082-989X.11.1.54>). Similarly, for the rate of grooming, a more appropriate model is one with a Poisson error structure that includes an offset (use 'offset' in R) of the exposure time (the time available to groom, log transformed) as an additional fixed effect. Using an offset in a Poisson model is a conventional way of analysing rate data (See Crawley MJ. 2007. *The R book*, Chapter 9 of Zuur et al. 2009. *Mixed effects models and extensions in ecology with R*, and numerous online forums).

3. I have concerns with the stepwise elimination of fixed effects conducted as part of statistical analyses to obtain a minimal model. The use of stepwise elimination of non-significant terms (specifically main effects) is widely accepted that this method of model simplification is flawed because it often results in overestimated effect sizes, multiple hypothesis testing and inflated type I error rates. See these papers for discussion:

Forstmeier, W., & Schielzeth, H. (2011). Cryptic multiple hypotheses testing in linear models: Overestimated effect sizes and the winner's curse. *Behavioral Ecology and Sociobiology*, 65(1), 47e55. <http://dx.doi.org/10.1007/s00265-010-1038-5>.

Mundry, R., & Nunn, C. L. (2009). Stepwise model fitting and statistical inference: Turning noise into signal pollution. *American Naturalist*, 173(1), 119e123. <http://dx.doi.org/10.1086/593303>.

Whittingham, M. J., Stephens, P. A., Bradbury, R. B., & Freckleton, R. P. (2006). *Journal of Animal Ecology*, 75(5), 1182e1189.

While removing non-significant interaction terms in order to test main effects is accepted (see Engquist (2005) *Animal Behaviour* 70, 967-971 doi:10.1016/j.anbehav.2005.01.016), removing non-significant main effects changes the partitioning of variance between the remaining main effects and so can lead to the problems I have outlined above. My recommendation would be to retain non-significant main effects in the model and report the estimates, standard errors, test statistics and p-values from these full models.

Other comments:

4. Title and throughout manuscript: I am not sure that the use of the term 'carryover' to capture the effects observed the day after exposure to a rival threat is the most appropriate. Carryover effects in ecology are usually used for effects observed in one breeding season or part of the life history that have been influenced by the organism's experience in the breeding season (or part of life history) before. Would 'longer term' or 'delayed' effects in the context of this study be more appropriate?

5. L45: see also Preston et al. 2020 *Journal of Animal Ecology*, 90 153-167 DOI: 10.1111/1365-2656.13323 for a study that examines longer term consequences of simulated intergroup conflict on grooming behaviour.

6. L119-126: Can you give directional predictions alongside the rationale, rather than just saying that behavioural responses will be 'greater' following multiple exposures, or for certain individuals. Also, it would be nice to include predictions of the effect on rarer group latrining and fissioning events. At the moment these seem like a bit of an afterthought because these behaviours were able to be observed rather than forming part of the set up of the study.

7. L142: I'm not sure I understand the statement that the playback was not used for data collection purposes. Why use the playback at all?

8. L153-154: What is the rationale for using a non-neighbouring group as the rival threat, rather than a rival group that shared a territorial boundary with the focal group? Does a non-neighbour represent a larger threat (i.e. a dear enemy effect)?
9. L162: 'a playback' or 'playbacks'.
10. L175: Can you provide mean duration (and range) between trials?
11. L187-190: Can you provide means +/- SE for the treatments in the results for Wilcoxon tests so that the reader can more easily evaluate the strength of the effect.
12. L201-203: How confident can you be that the observer was able to record all interactions and behaviours throughout the day as (potentially) the only observer while the group was foraging and potentially moving in and out of sight?
13. L226-227: Can you provide more information on how P-values were generated using the Monte Carlo resampling method? This seems like an important part of analyses and subsequent interpretation of data (P-values are used to determine significance) and so more detail is needed on how this was performed.
14. L230-231: More information is required on the models in the main text of the manuscript, rather than relying on this information being accessed in the ESM. The minimum amount of information provided on statistical models should be the response variable, the type of model (e.g. the error structure used), and the fixed and random effects. Without this information it is difficult to evaluate the results and the subsequent interpretation. To me, this information is critical to the paper and should be provided to the reader in the main text. In general I found the statistical methods not particularly well explained and quite difficult to follow because the information is scattered throughout the main text, ESM and ESM table headings.
15. L236 and throughout: Include model estimates and SEs with statistical results so that the reader can more easily evaluate the strength of the effect. The test statistic and P-value alone is not sufficient to assess effect size which is arguably a more informative metric than the P-value.
16. L242 and throughout: Use 'status' rather than 'dominance' when referring to the dominance status variable to be consistent with terminology in the ESM.
17. L242: The stats report here are different to the ones in Table S2a in the ESM.
18. L250: I think you mean the probability of the group foraging at each scan sample.? As I understand, this is what you tested in your binomial model; the proportion of scans that the group was observed foraging is a subtly different metric (as is the amount of foraging).
19. L251: As above, do you mean the probability of the group sentinelling at each scan?
20. L249-252 & L253-256: I am very surprised that you have been able to run a GLMM with only 14 data points for these analyses, particularly with a number of fixed and random effects. Additionally, you don't provide any estimates for the effects that you are reporting here, even in the ESM (perhaps because you remove them from the model?). This is important information to report - non-significant effects will have an estimate from the model and if you are reporting, and particularly plotting, these results then this information should be included.
21. L329-331 & L347-349: I am not convinced that you have shown that the effect on day 6 is stronger than that on day 2 (see my main comment 1 above).
22. L356-360: Do you know with whom males/subs are grooming? Can you comment on whether there might be some consolidating of bonds between, e.g. subs and subs who might be more involved in an intergroup interaction?
23. L366-367: Might an appropriate reference here be Thompson & Cant 2018 Behav Ecol doi:10.1093/beheco/ary044
24. L369-371: What is the frequency of naturally occurring intergroup interactions in this species, and how relevant is it to experience a barrage of threat stimuli in a short space of time? How might the response of dwarf mongooses to repeated stimuli compare to other species in which intergroup fighting takes place very often, and is more intensely aggressive (e.g. banded mongooses, meerkats)? Perhaps the strength of the response in dwarf mongooses might be larger because rivals are not encountered as frequently? Including some information on the frequency of intergroup interactions somewhere in the manuscript (possibly when describing the system) might help to further contextualise the results.
25. All figures: Could you also plot the predicted means from your models as well as the raw data. Currently there are a number of plots where there is a statistically significant difference between treatments, but the difference between the raw data does not look very convincing (e.g.

Fig 1b – there does not appear to be much difference in the relationships of doms and subs across control and intrusion trials; Fig 2c day 6 panel). Plotting the estimates (with standard error) from your models, alongside raw data, might allow statistically significant differences between categories to be more apparent because the estimated mean will be different from the median of the boxplots that are currently presented.

26. Fig 1: If an individual only groomed in one treatment then they will have a matching point at zero for the treatment in which they did not groom (i.e. there won't be any orphan points; data points at zero are still data points and should not be removed). Do you mean that some individuals were only observed in one trial and therefore only contributed one data point?

27. Fig 2a: y-axis should be probability of group foraging rather than proportion of group foraging activity? Similarly for Fig 2b, probability not proportion?

28. Fig 4: I am not able to marry these figures up with descriptions of the analysis because there is little (no) description of the way in which these rare events were analysed, either in the main text or the ESM.

29. ESM: In the Data Analysis section, I would like it explained explicitly what fixed and random effects were included in each model with some rationale as to why. Currently, I am unclear as to what variables were included in analyses and have to refer to ESM tables to find out. I think this information is better presented in the main text (see my comment above).

30. ESM: In the data analysis section, you describe sequentially removing terms using drop1 but drop1 does not remove the terms, it tests the effect of removing each fixed effect in turn against the full model containing that variable. This is what you describe as the 'ANOVA model comparison', but at the moment it sounds like drop1 is used to remove effects (rather than decide which effects to remove). I have my objections to stepwise elimination of variables to simplify models, which I have already outlined.

31. It is not clear why you refer to using Akaike Information Criteria to evaluate fixed effects when there is no reporting of any AIC values for any models. Throughout the study, P-values are used to determine the significance of effects and so to also use AIC values conflates two very distinct statistical approaches and philosophies. Can you remove reference to AIC values to avoid confusion?

32. ESM: I do not understand this sentence: "We added non-significant effects individually back to the minimal model to obtain significance levels, whilst we obtained values for significant effects either by comparing the term with a null model or by comparing the full minimal model with each term removed individually." What do you mean by 'values for significant effects'?

33. ESM: Why do you round nearest neighbour distances that are less than 0.5m to 0.1m rather than rounding to 0?

34. ESM: When calculating body mass change, might you have a problem with regression to the mean whereby extremely high or low values in the first measure of a given variable are more likely to move closer to the mean in a second measure of that variable. You can control for regression to the mean (e.g. Kelly & Price 2005 *Am Nat* 166, 700-707 doi:10.1086/497402).

35. ESM Tables: you should give details of any data transformations in the methods (ideally of the main text) rather than burying this information in an ESM table heading.

36. ESM Tables: What are your reference levels for sex and dominance status?

37. ESM Tables: Please report sample size with statistical tables.

38. ESM Tables: Why are effect sizes and standard errors for removed variables missing? This is important information, particularly when these patterns are reported and plotted in the manuscript, and so needs to be included.

Referee: 2

Comments to the Author(s)

This manuscript is another fascinating and well thought through and implemented experiment from the Radford group that can be used as a showcase of how to conduct experiments. Here Amy Morris-Drake and colleagues investigate the behavioural long-term effects of inter-group conflict using an excellent paradigm to investigate whether or not this effect is different after a single conflict or after the accumulation of several conflicts. They find several behavioural differences, some of which change due to a likely cumulative effect. The data and the results are convincing, although some questions remain regarding to the statistical analysis, which to my

understanding need some improvement. Their conclusions are driven by the results and explained in easy understandable and well structured sentences. Overall, this paper is a very good example on how such a topic can be researched and such questions can be addressed by an experiment.

I have three major points with regards to the statistical analysis:

(1) with small data sets like this (N=7) exact tests should be conducted. This is important because distributions with such a small sample size are not easy to assess. Please check Mundry & Fischer 1998 in *Animal Behaviour*

(2) the authors seem to undergo a model selection process removing non-significant fixed effects. While this is usual procedure for non significant interactions of fixed effects, this is seen as multiple testing when done with fixed effects, whether they are test or control predictors. I suggest to follow a similar procedure like in Wegdell et al. 2019 for an experiment in Guinea baboons.

(3) an eloquent way of checking whether or not your predictors of interest are working well is to test explanatory power in a full-null model comparison. The authors have some clear predictions and should therefore do so. This makes a complex model very simple reduced to one test. Here the authors could also follow the methods described in Wegdell et al. 2019

I believe that the results are very robust and most of the results will stand with this statistical method as well. The advantage of this method is that the results will not be seen as stronger and more rigorous.

I have to state clearly that this paper is truly an advance in the study of inter-group conflict and that I really enjoyed reading and thinking about it. I fully support its publication.

Specific comments:

line 86-87: I am not entirely sure what the argument here is: it is true that a cumulative effect is the result of shorter latency of conflicts - but by definition this is also what an experimental approach is doing - it is increasing the frequency of inter-group conflict by adding additional conflicts to the naturally occurring ones. The advantage of the experiment on the other side is that the amount and the numbers can be obviously controlled - see for example Wilson et al 2001, Herbinger et al. 2009

line 129 - 149: A figure of the experimental protocol would be good, since it is a bit confusing how often and for which duration the test (intrusion) and the control stimuli have been presented: (in the beginning it is on day 1-5 of the week, over two weeks, then in the control it is suddenly only one week - talking about the control week, and later (line 140) there is a range of 3-5 days of feacal presentation. In addition there seems to be a variation of playback experiments that have been conducted per group, potentially creating a different type of threat (immediate) compared to the feacale presentation.

line 194: DMRP = ?

line 347 - 350: It would be interesting to see if the groups had changed their ranging behaviour as well. I assume that the burrow of mangose is more or less in the middle of their territory and having signs of a stranger group at the burrow, in the center of the territory, must be more frightening than in the periphery. So after a week of reoccurring sign of competitors at the own burrow - are they more likely to stay close to the burrow, to not loose it suddenly to unwanted guests?

Author's Response to Decision Letter for (RSPB-2021-1743.R0)

See Appendix A.

Decision letter (RSPB-2021-1743.R1)

11-Nov-2021

Dear Professor Radford:

Your manuscript has now been peer reviewed and the reviews have been assessed by an Associate Editor. The reviewers' comments (not including confidential comments to the Editor) and the comments from the Associate Editor are included at the end of this email for your reference. As you will see, the reviewers and the Editors have raised some concerns with your manuscript and we would like to invite you to revise your manuscript to address them.

Research ethics:

Use of animals and field studies:

It is a condition of publication that you make available the data and research materials supporting the results in the article (<https://royalsociety.org/journals/authors/author-guidelines/#data>). Datasets should be deposited in an appropriate publicly available repository and details of the associated accession number, link or DOI to the datasets must be included in the Data Accessibility section of the article (<https://royalsociety.org/journals/ethics->

policies/data-sharing-mining/). Reference(s) to datasets should also be included in the reference list of the article with DOIs (where available).

Please submit a copy of your revised paper within three weeks. If we do not hear from you within this time your manuscript will be rejected. If you are unable to meet this deadline please let us know as soon as possible, as we may be able to grant a short extension.

Best wishes,
Dr Locke Rowe
Editor, Proceedings B
mailto: proceedingsb@royalsociety.org

Associate Editor

Comments to Author:

Thank you for your careful responses to the reviewer comments – it is much appreciated. I hope the authors agree that the paper has been strengthened as a result of these changes. There is one particular aspect to which the authors have only responded in part and in my mind, still needs to be addressed. It is the point raised by Reviewer 1 about the direct comparison of Day 2 and Day 6 grooming behavior. While the body mass aspect related to this was addressed in the revision, the authors did not directly compare Day 2 and Day 6 in grooming behavior, suggesting that the current separate analyses showing a significant treatment effect on Day 6 but not Day 2 adequately demonstrates a change through time. Unfortunately, I disagree with this interpretation. Importantly, when examining proportion of time spent grooming (or other patterns), Day 2 indeed does not show a significant main treatment effect but does show a significant treatment*sex interaction. This significant interaction means that the main effect of treatment can be misleading – specifically, that the effect of treatment depends on the sex

examined. As we can see from figure 2, looking at males suggests a strong treatment effect, while in females this is less so. This is super interesting but it does make comparisons of main treatment effects across analyses not appropriate. Furthermore, the models conducted on Day 2 and 6 are slightly different (one having a logit-link function and the other a probit-link function; line 251-252) making it even more difficult to make a comparison. I encourage the authors to examine the Day 2 vs Day 6 patterns directly with separate analyses that build on the current ones. Specifically, this would involve examining intrusion weeks only and including the effect of time (day 2 vs day 6) along with the other factors. I understand that this results in an increased number of tests so I would encourage the authors to only explore this time-aspect for behaviors in which patterns seem to be changing (as follow-ups). In this way, the conclusions about patterns becoming "more prominent on Day 6 than Day 2" could be supported with direct evidence. I should add that the results of these analyses would not affect the impact of this paper and would simply affect certain conclusions (line 342-345, 391-393, 409-411).

I would also suggest that the authors include the number of individuals along with the group number in their sample size numbers throughout (e.g. 47 individuals from 7 groups). Showing an $N=7$, while correct for this type of repeated design, does not provide any information about the number of individuals on which the behavior was actually measured.

This is a solid, novel study with great appeal to a broad audience and I look forward to seeing a revised version of this study soon.

Sincerely,
Katie McGhee
Associate Editor

Author's Response to Decision Letter for (RSPB-2021-1743.R1)

See Appendix B.

Decision letter (RSPB-2021-1743.R2)

17-Nov-2021

Dear Professor Radford

I am pleased to inform you that your manuscript entitled "Extended and cumulative effects of experimentally induced intergroup conflict in a cooperatively breeding mammal" has been accepted for publication in Proceedings B.

Data Accessibility section

Open Access

Paper charges

Sincerely,

Dr Locke Rowe

Associate Editor:

Board Member

Comments to Author:

Thank you! It all looks good!

Best,

Katie McGhee

Appendix A

Extended and cumulative effects of experimentally induced intergroup conflict in a cooperatively breeding mammal

Amy Morris-Drake, Jennifer F. Linden, Julie M. Kern & Andrew N. Radford

Response to Reviewers

We are grateful to the two reviewers and the Associate Editor for their positive and constructive thoughts about our work and manuscript. In taking those into account (we provide our responses in bold beneath each comment, with line numbers referring to highlighted sections in the revised manuscript), we believe that we have strengthened the paper further.

Associate Editor:

I have now received thorough reviews from two experts. Both reviewers were positive about the topic and complimentary on the methods and quality of the manuscript. The concerns raised were associated with the statistical analyses and additional clarification of methods. From my own reading of the manuscript, I am in agreement with the reviewer comments and concerns.

Reviewer 2 raises an important point about inferring differences between time periods based on analyses within time periods, rather than directly assessing the differences. Their suggestions are reasonable and should allow you to directly test whether there are indeed differences between day 2 and day 6. This is necessary for any comparison among times.

We have included a direct comparison of Day 2 and Day 6; full details in response to Reviewer 1 point 1.

Both reviewers commented on the potential issues associated with the removal of nonsignificant fixed effects. I am in agreement here that these fixed effects should not be removed (although nonsignificant interactions with them can be). In addition to the statistical concerns raised, these fixed effects are presumably biologically relevant to the study (thus their initial inclusion) and were part of the original design. Their inclusion also allows for comparison of their influence across behaviors and times in a more standard way.

We have redone our mixed models to avoid this issue (i.e., we have no longer removed non-significant fixed effects; full details in response to Reviewer 1 point 3). All our major conclusions remain as in the original submission.

Reviewer 1 suggests using different methods potentially better suited to the grooming data and these are well-worth exploring. If patterns are consistent across methods and the transformations resulted in well-behaved residuals in the original analyses, I do not feel strongly about needed to replace the analyses. However, it would be helpful to include these additional methods in the ESM if you decide to retain the original analyses.

We have redone these analyses of grooming data as suggested (full details in response to Reviewer 1 point 2). The major conclusions remain as in the original submission.

Reviewer 1 suggests a number of areas in the methods and results that need further clarification or information. I am also in agreement with reviewer 2 that a figure illustrating the experimental design would be very helpful – perhaps a smaller and less detailed version of Figure S1 in the ESM? Additionally, information defining ‘close’ and ‘lost’ calls from the ESM should be included in the main text (line 162).

We have provided additional clarification where requested, added a simplified figure of the experimental design to the main text, and moved information on calls from the ESM (full details in response to individual comments below).

Referee 1:

This study investigates the effects of simulated intergroup conflict on within-group affiliative and social behaviour using field experiments in a population of wild dwarf mongooses (*Helogale parvula*). The authors test whether exposure to a single rival group stimulus leads to longer-term changes in grooming, foraging and vigilance behaviour, and weight gain (over the course of the next day), and whether repeated exposure to rival group stimuli results in a cumulative effect on within-group behaviour. They find evidence for both 'carryover' and cumulative effects in groups and, interestingly, that behavioural responses are different for different individuals (dominants versus subordinates, males versus females). They conclude that intergroup threat has the potential to serve as a strong selective force shaping within-group behaviour and fitness.

I congratulate the authors on designing and performing a neat experiment with appropriate and well thought out controls – a great effort in the field where so much can go wrong, particularly when conducting a week-long trial. While sample sizes are not huge (particularly for the group-level metrics), the value of these types of experimental studies are notable and the results can provide an important and interesting contribution to the field when considered carefully. I found the paper well written and enjoyable to read.

Thank you for all these positive comments; they are much appreciated.

Main comments:

1. The test of cumulative effects of repeated exposure to simulated threat on day 6 does not directly test for cumulative effects because there is no direct statistical comparison between day 2 behaviour and day 6 behaviour (only day 6 Intrusion trials to day 6 Control trials). This makes any conclusions about there being a cumulative effect difficult to make, and therefore statements of there being a "stronger treatment difference on day 6 than day 2" (e.g. L329-331, L347-349) are unsupported. Currently, the results show that mongooses respond to repeated exposures to threats (as compared to repeated exposure to a benign stimulus), but not that this response is stronger than after just one exposure (as is claimed by a observed cumulative effect). I believe it is possible to test for a cumulative threat by analysing day 2 and day 6 data from Intrusion trials in one analysis (for each response variable), and including time as the explanatory variable of interest (perhaps also including interactions between time and dominance status or sex to investigate inter-individual responses as per the original analyses). Alternatively, day 2 and day 6 data for both Intrusion and Control trials could be included in one analysis (for each response variable) and the interaction between time and treatment used to test whether the behavioural response on day 6 is different to day 2 (for Intrusion trials, as predicted, but not for Control trials). The potential problem with this approach is that testing for inter-individual differences would involve testing 3-way interactions (e.g. time*treatment*sex) and 3-way interactions are complicated and tricky to interpret. My recommendation would be to include an analysis in addition to those already performed that explicitly tests data from Intrusion trials for day 2 and day 6, as I described. This would mean that claims of a cumulative effect can be supported statistically, rather than inferred from two separate analyses (and plots of data).

We do believe that there is value in comparing the findings on Day 2 with those on Day 6: when there is no treatment difference on Day 2 but a treatment difference on Day 6 (as is the case for our foraging and body-mass results, for instance), this indicates that something has indeed changed during the week. However, we recognise that care needs to be taken regarding the exact nature of the resulting conclusions – we have therefore made sure that the wording related to such comparisons does not mention 'cumulative' explicitly (e.g., lines 20–21, 248–249, 342–343).

So, we have included direct comparisons of the change from Day 2 to Day 6 (as suggested) for the body-mass data (lines 278–282). The reason for focussing on those datasets is that body-mass changes likely reflect the collective effect of behavioural changes (lines 126–127, 270–271), and they are also most directly relevant to fitness consequences (lines 363–368). [It is for this reason, and to avoid the group-level behaviours feeling 'tacked on', that we now finish our Results with the body-mass analyses.] We have compared between treatments, rather than considering only Intrusion weeks, because that is in-line with our overall experimental design. These new analyses show that there is indeed a significantly greater difference in the body-mass effect across Intrusion weeks cf. Control weeks (lines 330–333).

We believe that together, a combination of our results – no treatment difference on Day 2 but a significant difference on Day 6 for some response measures, the treatment differences across the week for group latrining and fissioning, and the new direct comparison of change from Day 2 to Day 6 for body-mass gain – allow the overall conclusion about a cumulative effect; this is now how we have worded our Discussion summary on this aspect of the study (lines 342–351).

2. The types of models (transformations of data in a model with a Gaussian error structure) used to analyse the proportion of time grooming and the rate of grooming in linear models is not the most appropriate given the distribution of the data. Neither of these variables are normally distributed by definition (proportion time is bounded by 0 and 1, and a rate is a count of occurrences per unit time). A more appropriate model to use for the proportion time data is one with a beta error distribution which accommodates bounded variables (e.g. Smithson & Verkuilen 2006 <https://doi.org/10.1037/1082-989X.11.1.54>). Similarly, for the rate of grooming, a more appropriate model is one with a Poisson error structure that includes an offset (use 'offset' in R) of the exposure time (the time available to groom, log transformed) as an additional fixed effect. Using an offset in a Poisson model is a conventional way of analysing rate data (See Crawley MJ. 2007. The R book, Chapter 9 of Zuur et al. 2009. Mixed effects models and extensions in ecology with R, and numerous online forums).

We have reanalysed these two response variables using the suggested methods (lines 249–256). We find qualitatively similar results from before – i.e., there is already some effect on grooming on Day 2, but the amount of inter-individual variation increases by Day 6 (line 285ff.).

3. I have concerns with the stepwise elimination of fixed effects conducted as part of statistical analyses to obtain a minimal model. The use of stepwise elimination of non-significant terms (specifically main effects) is widely accepted that this method of model simplification is flawed because it often results in overestimated effect sizes, multiple hypothesis testing and inflated type I error rates. See these papers for discussion:

Forstmeier, W., & Schielzeth, H. (2011). Cryptic multiple hypotheses testing in linear models: Overestimated effect sizes and the winner's curse. *Behavioral Ecology and Sociobiology*, 65(1), 47e55. <http://dx.doi.org/10.1007/s00265-010-1038-5>.

Mundry, R., & Nunn, C. L. (2009). Stepwise model fitting and statistical inference: Turning noise into signal pollution. *American Naturalist*, 173(1), 119e123. [http:// dx.doi.org/10.1086/593303](http://dx.doi.org/10.1086/593303).

Whittingham, M. J., Stephens, P. A., Bradbury, R. B., & Freckleton, R. P. (2006). *Journal of Animal Ecology*, 75(5), 1182e1189.

While removing non-significant interaction terms in order to test main effects is accepted (see Engquist (2005) *Animal Behaviour* 70, 967–971 doi:10.1016/j.anbehav.2005.01.016), removing non-significant main effects changes the partitioning of variance between the remaining main effects and so can lead to the problems I have outlined above. My recommendation would be to retain non-significant main effects in the model and report the estimates, standard errors, test statistics and p-values from these full models.

We have redone our mixed models without stepwise elimination of non-significant main effects (lines S138–S146) and we have reported the estimates and standard errors from these maximal models (Tables S1–S6). We have also assessed the significance of key terms in the maximal model using likelihood ratio tests, comparing the maximal model to a model without the term of interest (lines S146–S149); these values are also reported in Tables S1–S6. We find qualitatively the same results as before, with the minor exception that the previous sentinel result that showed a non-significant trend on Day 6 is now non-significant, so all main conclusions are as in the original submission.

Other comments:

4. Title and throughout manuscript: I am not sure that the use of the term 'carryover' to capture the effects observed the day after exposure to a rival threat is the most appropriate. Carryover effects in ecology are usually used for effects observed in one breeding season or part of the life history that have been

influenced by the organism's experience in the breeding season (or part of life history) before. Would 'longer term' or 'delayed' effects in the context of this study be more appropriate?

We have removed use of 'carryover' and replaced it with 'extended' throughout the manuscript (e.g., lines 27, 46, 66 and onwards), including in the title. We have chosen to use 'extended' because 'longer-term' is a comparator, so we would need to keep saying what it was longer than, and 'delayed' might create the wrong impression that there is only a later response and no initial one (which previous work demonstrates there is).

5. L45: see also Preston et al. 2020 Journal of Animal Ecology, 90 153-167 DOI: 10.1111/1365-2656.13323 for a study that examines longer term consequences of simulated intergroup conflict on grooming behaviour.

We have added this reference (line 45).

6. L119-126: Can you give directional predictions alongside the rationale, rather than just saying that behavioural responses will be 'greater' following multiple exposures, or for certain individuals. Also, it would be nice to include predictions of the effect on rarer group latrining and fissioning events. At the moment these seem like a bit of an afterthought because these behaviours were able to be observed rather than forming part of the set up of the study.

We have added directional predictions, including for the group latrining and fissioning events (lines 123–127). To help avoid the feeling that the rarer group events were an afterthought (which they weren't), we have integrated them better (in the Introduction and Results) – including them after the individual-level behaviours and finishing with the body-mass change element (as that likely reflects the collective result of various behavioural changes; see above).

7. L142: I'm not sure I understand the statement that the playback was not used for data collection purposes. Why use the playback at all?

We apologise for the confusion created by our previous wording. The playback was to maintain rival-group or herbivore exposure without constantly needing faecal samples. We have removed mention of 'data collection' (lines 147–148).

8. L153-154: What is the rationale for using a non-neighbouring group as the rival threat, rather than a rival group that shared a territorial boundary with the focal group? Does a non-neighbour represent a larger threat (i.e. a dear enemy effect)?

Non-neighbouring groups are encountered naturally (lines 140–142) and we have added a rationale for choosing to use them as the simulated threat in this experiment (lines S25–S29). Earlier work we conducted on the study population, looking at immediate changes in latrining and movement behaviour, found no significant difference in response to neighbours and non-neighbours (Christensen et al. 2016). We therefore chose to use non-neighbours in the current work to be consistent (and allow direct comparison) with an earlier experiment we published looking at short-term behavioural responses (i.e., those in the hour after a single presentation) (Morris-Drake et al. 2019).

9. L162: 'a playback' or 'playbacks'.

We have changed to 'playbacks' (line 169).

10. L175: Can you provide mean duration (and range) between trials?

We have included this information (line 186).

11. L187-190: Can you provide means +/- SE for the treatments in the results for Wilcoxon tests so that the reader can more easily evaluate the strength of the effect.

We have included these summary values for the treatments (lines 198, 200, 202–203).

12. L201-203: How confident can you be that the observer was able to record all interactions and behaviours throughout the day as (potentially) the only observer while the group was foraging and potentially moving in and out of sight?

We cannot claim never to have missed occurrences of interactions or behaviours, but there is no a priori reason to expect there to be any systematic bias between treatments or days of the week in which group members were visible.

13. L226-227: Can you provide more information on how P-values were generated using the Monte Carlo resampling method? This seems like an important part of analyses and subsequent interpretation of data (P-values are used to determine significance) and so more detail is needed on how this was performed.

In response to a comment by Reviewer 2, we have changed our methodology to use exact testing (line 241) as opposed to Monte Carlo resampling for the generation of P-values; all results are qualitatively the same as before.

14. L230-231: More information is required on the models in the main text of the manuscript, rather than relying on this information being accessed in the ESM. The minimum amount of information provided on statistical models should be the response variable, the type of model (e.g. the error structure used), and the fixed and random effects. Without this information it is difficult to evaluate the results and the subsequent interpretation. To me, this information is critical to the paper and should be provided to the reader in the main text. In general I found the statistical methods not particularly well explained and quite difficult to follow because the information is scattered throughout the main text, ESM and ESM table headings.

We have included more detailed information (i.e., response variable, type of model, fixed and random terms) in the main text (lines 248–282).

15. L236 and throughout: Include model estimates and SEs with statistical results so that the reader can more easily evaluate the strength of the effect. The test statistic and P-value alone is not sufficient to assess effect size which is arguably a more informative metric than the P-value.

We have included this additional information in the main text (e.g., lines 287, 289, 291 and onwards), as well as in the Tables containing full statistical output (Tables S1–S6).

16. L242 and throughout: Use 'status' rather than 'dominance' when referring to the dominance status variable to be consistent with terminology in the ESM.

We have changed 'dominance' to 'status' in the main text to be consistent with the ESM (line 299).

17. L242: The stats report here are different to the ones in Table S2a in the ESM.

Statistical output values have changed with the remodelling (see earlier responses), but we have carefully checked throughout to make sure that those presented in the main text exactly match those in the Tables.

18. L250: I think you mean the probability of the group foraging at each scan sample.? As I understand, this is what you tested in your binomial model; the proportion of scans that the group was observed foraging is a subtly different metric (as is the amount of foraging).

This is the proportion of scans, so we have left the wording as is (line 303). As a group-level response, we analysed this using a Wilcoxon signed-ranks test (lines 240–241), comparing for each group the proportion of scans where foraging was recorded in each of the two treatments.

19. L251: As above, do you mean the probability of the group sentinelling at each scan?

Here, we were analysing individual behaviour and so did indeed assess whether a sentinel was present in each scan conducted; as such, we have changed the wording accordingly (lines 261–262, 304–305).

20. L249-252 & L253-256: I am very surprised that you have been able to run a GLMM with only 14 data points for these analyses, particularly with a number of fixed and random effects. Additionally, you don't provide any estimates for the effects that you are reporting here, even in the ESM (perhaps because you remove them from the model?). This is important information to report – non-significant effects will have an estimate from the model and if you are reporting, and particularly plotting, these results then this information should be included.

These mixed-model analyses have not been conducted with 14 datapoints (i.e., one value per group per treatment). The sentinel analyses use each scan-sample as a data point, with the presence or absence of a sentinel as the binomial response term (lines 261–262). The nearest-neighbour analyses use each individual's nearest-neighbour distance in each scan sample as a data point (lines 264–265). Our additional information about each model, provided in one place in the main text (in response to an earlier comment) will hopefully assist understanding of exactly what was done in each case.

21. L329-331 & L347-349: I am not convinced that you have shown that the effect on day 6 is stronger than that on day 2 (see my main comment 1 above).

We would argue that if there is no significant treatment effect on Day 2, but there is a significant treatment effect on Day 6, that does demonstrate a 'stronger' effect (especially where the sample sizes for the two days are equivalent). However, we have also included some additional analyses assessing the difference between Day 2 and 6 directly (see response to comment 2 earlier).

22. L356-360: Do you know with whom males/subs are grooming? Can you comment on whether there might be some consolidating of bonds between, e.g. subs and subs who might be more involved in an intergroup interaction?

Unfortunately, we don't have large enough sample sizes of dyadic interactions to be able to assess this formally; future work will be needed to get at the exact details in this respect.

23. L366-367: Might an appropriate reference here be Thompson & Cant 2018 Behav Ecol doi:10.1093/beheco/ary044

We have added this reference (line 428).

24. L369-371: What is the frequency of naturally occurring intergroup interactions in this species, and how relevant is it to experience a barrage of threat stimuli in a short space of time? How might the response of dwarf mongooses to repeated stimuli compare to other species in which intergroup fighting takes place very often, and is more intensely aggressive (e.g. banded mongooses, meerkats)? Perhaps the strength of the response in dwarf mongooses might be larger because rivals are not encountered as frequently? Including some information on the frequency of intergroup interactions somewhere in the manuscript (possibly when describing the system) might help to further contextualise the results.

Whilst interactions between rival groups are not that common, at least compared to some other species, the encountering of secondary cues of recent rival-group presence is more frequent (due to overlap in space and shared latrine sites); sleeping burrows are regularly contested between groups. We have provided this additional information with other background on the study population (lines S20–S23) and when we explain the experimental regime (lines 187–188). As more research is conducted on the consequences of intergroup interactions, interspecific comparative studies will become more feasible... and they are certainly interesting as many factors can differ (e.g., not just the frequency of interactions, but their intensity/likelihood of escalation to physical fights).

25. All figures: Could you also plot the predicted means from your models as well as the raw data. Currently there are a number of plots where there is a statistically significant difference between treatments, but the difference between the raw data does not look very convincing (e.g. Fig 1b – there does not appear to be much difference in the relationships of doms and subs across control and intrusion trials; Fig 2c day 6

panel). Plotting the estimates (with standard error) from your models, alongside raw data, might allow statistically significant differences between categories to be more apparent because the estimated mean will be different from the median of the boxplots that are currently presented.

We have plotted predicted means as well as raw data in all figures relating to mixed models, as suggested.

26. Fig 1: If an individual only groomed in one treatment then they will have a matching point at zero for the treatment in which they did not groom (i.e. there won't be any orphan points; data points at zero are still data points and should not be removed). Do you mean that some individuals were only observed in one trial and therefore only contributed one data point?

Yes; we have reworded for clarity (lines 629–630).

27. Fig 2a: y-axis should be probability of group foraging rather than proportion of group foraging activity? Similarly for Fig 2b, probability not proportion?

See earlier response relating to this terminology.

28. Fig 4: I am not able to marry these figures up with descriptions of the analysis because there is little (no) description of the way in which these rare events were analysed, either in the main text or the ESM.

We have included more information on all our analyses and put it all in one place in the main text (lines 240–241).

29. ESM: In the Data Analysis section, I would like it explained explicitly what fixed and random effects were included in each model with some rationale as to why. Currently, I am unclear as to what variables were included in analyses and have to refer to ESM tables to find out. I think this information is better presented in the main text (see my comment above).

We have included more information about the fixed and random effects included in all our models in one place in the main text (lines 247–282).

30. ESM: In the data analysis section, you describe sequentially removing terms using drop1 but drop1 does not remove the terms, it tests the effect of removing each fixed effect in turn against the full model containing that variable. This is what you describe as the 'ANOVA model comparison', but at the moment it sounds like drop1 is used to remove effects (rather than decide which effects to remove). I have my objections to stepwise elimination of variables to simplify models, which I have already outlined.

We have redone all our models (as per main comments earlier); our description of what we did has therefore changed accordingly (lines S138–S149).

31. It is not clear why you refer to using Akaike Information Criteria to evaluate fixed effects when there is no reporting of any AIC values for any models. Throughout the study, P-values are used to determine the significance of effects and so to also use AIC values conflates two very distinct statistical approaches and philosophies. Can you remove reference to AIC values to avoid confusion?

We have redone all our models (see above) and this is no longer relevant so has been deleted.

32. ESM: I do not understand this sentence: "We added non-significant effects individually back to the minimal model to obtain significance levels, whilst we obtained values for significant effects either by comparing the term with a null model or by comparing the full minimal model with each term removed individually." What do you mean by 'values for significant effects'?

We have redone our models, so this is no longer relevant and has been deleted.

33. ESM: Why do you round nearest neighbour distances that are less than 0.5m to 0.1m rather than rounding to 0?

Because the mongooses were not physically touching.

34. ESM: When calculating body mass change, might you have a problem with regression to the mean whereby extremely high or low values in the first measure of a given variable are more likely to move closer to the mean in a second measure of that variable. You can control for regression to the mean (e.g. Kelly & Price 2005 Am Nat 166, 700-707 doi:10.1086/497402).

Kelly & Price (2005) state in their Introduction: 'Since regression to the mean will affect both experimental and control groups, a well-designed experimental study will not be subject to this problem'. The reviewers agree that ours is a well-designed study, so we do not believe that we have a problem with regression to the mean.

35. ESM Tables: you should give details of any data transformations in the methods (ideally of the main text) rather than burying this information in an ESM table heading.

We have provided all information about all models in one place in the main text (lines 247–282).

36. ESM Tables: What are your reference levels for sex and dominance status?

We have included reference levels in the table information.

37. ESM Tables: Please report sample size with statistical tables.

We have included sample sizes with all tables.

38. ESM Tables: Why are effect sizes and standard errors for removed variables missing? This is important information, particularly when these patterns are reported and plotted in the manuscript, and so needs to be included.

All relevant information for all variables is now included in the tables.

Referee: 2

This manuscript is another fascinating and well thought through and implemented experiment from the Radford group that can be used as a showcase of how to conduct experiments. Here Amy Morris-Drake and colleagues investigate the behavioural long-term effects of inter-group conflict using an excellent paradigm to investigate whether or not this effect is different after a single conflict or after the accumulation of several conflicts. They find several behavioural differences, some of which change due to a likely cumulative effect. The data and the results are convincing, although some questions remain regarding to the statistical analysis, which to my understanding need some improvement. Their conclusions are driven by the results and explained in easy understandable and well structured sentences. Overall, this paper is a very good example on how such a topic can be researched and such questions can be addressed by an experiment.

Many thanks for all these positive comments; it is much appreciated.

I have three major points with regards to the statistical analysis:

(1) with small data sets like this (N=7) exact tests should be conducted. This is important because distributions with such a small sample size are not easy to asses. Please check Mundry & Fischer 1998 in Animal Behaviour.

We have changed to use of exact testing as opposed to Monte Carlo resampling for the generation of p-values when conducting Wilcoxon signed-ranked tests (lines 240–241). No results have qualitatively changed as a consequence.

(2) the authors seem to undergo a model selection process removing non-significant fixed effects. While this is usual procedure for non significant interactions of fixed effects, this is seen as multiple testing when done with fixed effects, whether they are test or control predictors. I suggest to follow a similar procedure like in Wegdell et al. 2019 for an experiment in Guinea baboons.

We have altered our modelling to avoid this problem (full details in response to Reviewer 1 point 3).

(3) an eloquent way of checking whether or not your predictors of interest are working well is to test explanatory power in a full-null model comparison. The authors have some clear predictions and should therefore do so. This makes a complex model very simple reduced to one test. Here the authors could also follow the methods described in Wegdell et al. 2019.

In our new modelling approach (following the comments of both reviewers), we remove non-significant interaction terms from a full model to generate a maximal model (which contains significant interaction terms and all main effects). We present output values from this maximal model in our tables (lines S143–S146). For terms of interest (i.e., treatment and interactions with it), we also calculate p-values by comparing the maximal model with this model without the term of interest; these values are also presented in our tables (lines S146–S149). Thus, we already have two different assessments of the importance of our predictors of interest; we believe a third comparison would be overkill and, because there are several predictor variables in each model, a significant difference cf. a null model would not indicate which terms are important anyway.

Specific comments:

line 86-87: I am not entirely sure what the argument here is: it is true that a cumulative effect is the result of shorter latency of conflicts - but by definition this is also what an experimental approach is doing - it is increasing the frequency of inter-group conflict by adding additional conflicts to the naturally occurring ones. The advantage of the experiment on the other side is that the amount and the numbers can be obviously controlled - see for example Wilson et al 2001, Herbinger et al. 2009.

We apologise for the confusion created by our original wording. Our point was that, since we already know that there can be behavioural changes in the 1-2 hours after an intergroup interaction, if several of those take place in a given week, any overall change in behaviour in that week might simply be because there are several post-conflict periods. Whereas, the more interesting possibility (in our minds, at least) is that baseline behaviour, outside the immediate post-conflict periods, might also be changed due to cumulative effects. We have reworded to make this clearer (lines 86–91).

line 129 - 149: A figure of the experimental protocol would be good, since it is a bit confusing how often and for which duration the test (intrusion) and the control stimuli have been presented: (in the beginning it is on day 1-5 of the week, over two weeks, than in the control it is suddenly only one week - talking about the control week, and later (line 140) there is a range of 3-5 days of faecal presentation. In addition there seems to be a variation of playback experiments that have been conducted per group, potentially creating a different type of threat (immediate) compared to the faecale presentation. **We have included a figure of the experimental protocol in the main paper (rather than the ESM); it is the new Figure 1. Each trial was one week long (line 136), with exactly the same schedule for Intrusion and Control weeks (lines 142–143). There were multiple faecal presentations per week but always just the one playback (lines 145–147).**

line 194: DMRP = ?

Dwarf Mongoose Research Project. We had included the acronym with the full term on first mention. But since there are only two uses of it in the main MS, we have written it in full to avoid confusion (line 207).

line 347 - 350: It would be interesting to see if the groups had changed their ranging behaviour as well. I assume that the burrow of mangose is more or less in the middle of their territory and having signs of a stranger group at the burrow, in the center of the territory, must be more frightening than in the periphery. So after a week of reoccurring sign of competitors at the own burrow - are they more likely to stay close to the burrow, to not loose it suddenly to unwanted guests?

This is definitely an interesting question... but one for future work as it is a major undertaking to extract and analyse all those additional data. We are working on a more general ranging paper, which will include various factors that influence changes in space use, one of which is intergroup conflict.

Appendix B

Extended and cumulative effects of experimentally induced intergroup conflict in a cooperatively breeding mammal

Amy Morris-Drake, Jennifer F. Linden, Julie M. Kern & Andrew N. Radford

Response to Associate Editor

Comments to Author:

Thank you for your careful responses to the reviewer comments – it is much appreciated. I hope the authors agree that the paper has been strengthened as a result of these changes.

We are glad that the vast majority of our previous responses were satisfactory and absolutely agree that the paper has been strengthened as a result of the changes.

There is one particular aspect to which the authors have only responded in part and in my mind, still needs to be addressed. It is the point raised by Reviewer 1 about the direct comparison of Day 2 and Day 6 grooming behavior. While the body mass aspect related to this was addressed in the revision, the authors did not directly compare Day 2 and Day 6 in grooming behavior, suggesting that the current separate analyses showing a significant treatment effect on Day 6 but not Day 2 adequately demonstrates a change through time. Unfortunately, I disagree with this interpretation. Importantly, when examining proportion of time spent grooming (or other patterns), Day 2 indeed does not show a significant main treatment effect but does show a significant treatment*sex interaction. This significant interaction means that the main effect of treatment can be misleading – specifically, that the effect of treatment depends on the sex examined. As we can see from figure 2, looking at males suggests a strong treatment effect, while in females this is less so. This is super interesting but it does make comparisons of main treatment effects across analyses not appropriate. Furthermore, the models conducted on Day 2 and 6 are slightly different (one having a logit-link function and the other a probit-link function; line 251-252) making it even more difficult to make a comparison. I encourage the authors to examine the Day 2 vs Day 6 patterns directly with separate analyses that build on the current ones. Specifically, this would involve examining intrusion weeks only and including the effect of time (day 2 vs day 6) along with the other factors. I understand that this results in an increased number of tests so I would encourage the authors to only explore this time-aspect for behaviors in which patterns seem to be changing (as follow-ups). In this way, the conclusions about patterns becoming “more prominent on Day 6 than Day 2” could be supported with direct evidence. I should add that the results of these analyses would not affect the impact of this paper and would simply affect certain conclusions (line 342-345, 391-393, 409-411).

We have now conducted the additional grooming analyses exactly as suggested – i.e. models of Intrusion week data including day (Day 2, Day 6) and its interaction with both sex and dominance status (lines 259–268). Perhaps unsurprisingly, given that there are significant interaction terms in both the Day 2 and Day 6 grooming analyses (previously included results; lines 295–308), there were no significant differences in these new analyses (lines 308–311, full model outputs in Supplementary Table S3). This means we do not have clear evidence for our previous statements about inter-individual grooming differences being more pronounced on Day 6 than Day 2. As the Associate Editor states, this does not affect our major conclusions, but we have suitably tempered our wording in all appropriate places. I.e. we do not mention greater inter-individual grooming differences later in the week where we had previously done so in the Results (lines 306–308) and the Discussion (lines 354–356). In the Discussion paragraph about cumulative effects, we no longer include mention of grooming (line 401ff.). In the Discussion paragraph about inter-individual variation, we focus on the idea that this is apparent on both Day 2 and Day 6 (line 418ff.).

I would also suggest that the authors include the number of individuals along with the group number in their sample size numbers throughout (e.g. 47 individuals from 7 groups). Showing an N=7, while correct for this type of repeated design, does not provide any information about the number of individuals on which the behavior was actually measured.

Where we mention only the number of groups (i.e. N=7) as the sample size, this is because these analyses are of foraging activity, latrining and fissioning, which are group-level behaviours (lines 240–241); data were not collected from individuals. We have therefore left these sample sizes as they were. For all our individual-level responses (behaviour and body-mass change), we provide information on the number of individuals (as well as groups) in the Supplementary Tables and figure captions.

This is a solid, novel study with great appeal to a broad audience and I look forward to seeing a revised version of this study soon.

Thank you. We hope all is now sorted satisfactorily.